# Emotion-JEPA: Predictive Visual Adaptation and Audio-Modulated Fusion for Multimodal Emotion Recognition

## Abstract

Multimodal emotion recognition (MER) requires integrating visual, acoustic, and textual cues from short, noisy, and often ambiguous human expressions. Although pretrained encoders provide strong generic representations, MER remains challenging because affective cues are subtle, modality-dependent, and weakly supervised. We present Emotion-JEPA, a two-stage framework that studies affect-domain visual adaptation and structured audio-modulated fusion for MER. First, we adapt a V-JEPA2-style visual encoder on unlabeled emotion-domain videos using masked latent prediction, without pseudo-label generation or additional manual annotation. Second, we combine the adapted visual representation with audio and text through Audio-Modulated Hybrid Fusion (AMHF), which uses audio-conditioned grouped gating, sparse expert routing, memory-based refinement, learned confidence-normalized weighting, and residual fusion. On MER2024-SEMI, Emotion-JEPA achieves 85.72% weighted-average F1 (WAF), and replacing the adapted visual encoder with the public V-JEPA2 backbone reduces WAF by 7.92 points in the full multimodal pipeline. Controlled fusion comparisons show that AMHF improves over concat-MLP, cross-attention, and simpler audio-gated alternatives. Additional unimodal baselines, modulator variants, sparse-routing analysis, frozen visual probes, and a compute-bounded contrastive adaptation control clarify when adaptation and fusion contribute. The results support affect-domain V-JEPA2-style adaptation and structured audio-modulated fusion as useful design choices for MER under limited supervision.

## 1 Introduction

Multimodal emotion recognition (MER) aims to recognize affective expressions from visual, acoustic, and linguistic signals. This remains challenging because emotional cues are often subtle, short-lived, and unevenly expressed across modalities. Facial signals may be affected by pose, occlusion, or lighting, acoustic signals vary with speaker characteristics, background noise, and prosody, and transcripts may be sparse, incomplete, or only weakly related to the expressed emotion. These challenges are especially pronounced in limited-label settings, where models must learn to combine heterogeneous and noisy cues without over-relying on any single modality.

Large pretrained multimodal models provide strong general-purpose representations and have shown broad transfer across video, audio, and language tasks (Alayrac et al., 2022; Lin et al., 2024; Liu et al., 2023). However, direct transfer to MER remains difficult for two reasons. First, visual encoders trained on broad video corpora often emphasize scene semantics, object interactions, or action dynamics, whereas MER depends heavily on subtle facial motion, gaze changes, and short temporal expression cues. Second, standard fusion mechanisms, including cross-attention-based designs (Vaswani et al., 2017; Tsai et al., 2019), typically combine modalities without explicitly modeling how modality usefulness changes across samples. Consequently, improvements in MER can be difficult to attribute: they may arise from larger pretrained backbones, better in-domain representation alignment, or the inductive bias of the fusion mechanism itself.

This paper focuses on two factors central to this attribution problem: affect-domain visual adaptation and structured multimodal fusion. First, large video encoders provide strong generic representations, but

MER often depends on subtle facial motion, gaze changes, and short temporal affect cues that may not be emphasized by broad video pretraining. We therefore study whether in-domain adaptation of a V-JEPA2-style visual encoder on unlabeled emotion-domain videos improves MER when evaluated within a full multimodal pipeline. Second, because MER performance can be strongly influenced by unimodal encoder strength, we study audio-conditioned fusion under controlled encoder and training settings to better separate fusion gains from representation gains. Together, these questions motivate a controlled adaptation-and-fusion framework for MER under limited labeled supervision.

We introduce Emotion-JEPA, a two-stage framework for multimodal emotion recognition. Stage 1 adapts a V-JEPA2-style visual encoder using unlabeled emotion videos for masked latent prediction without pseudo-labels. Stage 2 combines the adapted visual encoder with pretrained audio and text encoders through Audio-Modulated Hybrid Fusion (AMHF), which uses acoustic evidence to guide visual and textual interaction through audio-conditioned modulation, sparse expert routing, memory-based refinement, learned confidence-normalized weighting, and progressive fusion. This design uses audio as a practical conditioning signal because acoustic prosody often provides useful affective evidence in MER, while visual and textual cues may vary in usefulness across samples.

We evaluate Emotion-JEPA on MER2024-SEMI under a controlled setting where unlabeled videos are used for self-supervised visual adaptation rather than pseudo-label generation. On MER2024-SEMI, Emotion-JEPA achieves 85.72% WAF, and removing the in-domain visual adaptation stage reduces WAF by 7.92 points in the full multimodal pipeline. Additional analyses compare AMHF with concat-MLP, cross-attention, simpler audio-gated fusion, alternative conditioning signals, and sparse-routing variants, while independent unimodal baselines and visual adaptation diagnostics help separate fusion gains from unimodal encoder strength. This scope emphasizes adaptation and fusion design rather than leaderboard optimization through pseudo-labeling, voting, or ensembling.

**Contributions:**

- **Affect-domain V-JEPA2-style visual adaptation for MER:** We adapt a V-JEPA2-style visual encoder on unlabeled emotion-domain videos using masked latent prediction, where unlabeled clips are used for self-supervised representation adaptation rather than pseudo-label generation. We evaluate this adaptation within the full multimodal pipeline and through visual-only diagnostics.

- **Audio-Modulated Hybrid Fusion for structured multimodal integration:** We introduce AMHF, a fusion module that uses audio-conditioned grouped gating, sparse expert routing, memory-based refinement, learned confidence-normalized weighting, and residual fusion to combine visual, acoustic, and textual evidence.

- **Controlled empirical analysis of adaptation and fusion:** We analyze Emotion-JEPA on MER2024-SEMI through framework ablations, strict fusion controls, independent unimodal baselines, modulator variants, sparse-routing sensitivity, visual adaptation diagnostics, and comparisons with representative MER systems and multimodal foundation models.

Together, these results support affect-domain visual adaptation and structured audio-modulated fusion as useful design choices for multimodal emotion recognition under limited supervision.

## 2 Related Work

### 2.1 Benchmarks and Limited Supervision in Multimodal Emotion Recognition

Recent progress in multimodal emotion recognition (MER) has been shaped by both benchmark design and model development. Early datasets such as IEMOCAP (Busso et al., 2008) enabled controlled studies of audiovisual and linguistic emotion recognition, while MERBench (Lian et al., 2026) highlighted the need for standardized evaluation across feature extractors, fusion modules, splits, and training protocols. The MER challenge series further expands this setting: MER2024 introduced semi-supervised learning, noise robustness, and open-vocabulary MER (Lian et al., 2024b), while MER2025 extends the benchmark toward

LMM-based affective computing, fine-grained recognition, and descriptive emotion understanding (Lian et al., 2025b).

Limited labeled data remains a central difficulty in this setting. Recent MER2024-SEMI systems often use pseudo-labeling, self-training, modality dropout, voting, or iterative refinement over unlabeled data (Qi et al., 2024; Ge et al., 2024; Shi & Gao, 2024). These strategies can improve benchmark performance by converting unlabeled examples into additional classification supervision. Emotion-JEPA also uses the unlabeled MER2024 clips, but uses them for self-supervised visual representation adaptation rather than pseudo-label generation. This distinction allows us to study affect-domain visual adaptation and audio-conditioned fusion under a controlled labeled-training protocol.

## 2.2 Large Multimodal Models for Affective Understanding

Large multimodal models (LMMs) such as CLIP (Radford et al., 2021), Flamingo (Alayrac et al., 2022), Video-LLaVA (Liu et al., 2023), Video-LLaMA (Zhang et al., 2023), and Qwen-VL/Qwen-Omni (Bai et al., 2023; Xu et al., 2025) provide strong general-purpose multimodal representations. Their success has motivated affective computing methods based on prompting, instruction tuning, and multimodal reasoning. For example, EmotionLLaMA (Cheng et al., 2024) adapts instruction-tuned multimodal models for emotion recognition and reasoning, while AffectGPT and MER-Caption extend this direction with descriptive emotion annotations and multimodal emotion understanding benchmarks (Lian et al., 2025a).

Recent work also explores open-vocabulary MER, where models predict flexible affective descriptions rather than fixed labels (Lian et al., 2024a). These developments show the growing role of LMMs in affective computing, but they leave open how modality-specific evidence should be adapted and fused in compact discriminative MER systems. Emotion-JEPA addresses this question by studying affective visual adaptation and audio-conditioned fusion under controlled encoder and training settings, rather than relying primarily on instruction tuning, generative descriptions, or model scale.

## 2.3 Self-Supervised Video Learning and Affective Adaptation

Self-supervised learning has become a standard approach for learning transferable visual representations, including contrastive methods (Chen et al., 2020; He et al., 2020), non-contrastive methods (Grill et al., 2020), and masked prediction models (He et al., 2022). In video understanding, predictive and masked objectives are especially relevant because they encourage models to capture temporal structure across frames.

The Joint-Embedding Predictive Architecture (JEPA) (Assran et al., 2023) predicts representations in latent space rather than reconstructing pixels. V-JEPA extends this idea to video through masked latent prediction (Bardes et al., 2024), and V-JEPA2 further scales predictive video representation learning (Assran et al., 2025). These models provide strong generic video features, but direct transfer may be suboptimal for MER because affective signals are often subtle, face-centric, and short-lived. Recent work has also explored V-JEPA for facial expression recognition using pretrained video encoders and shallow classifiers (Eing et al., 2026). Our work differs by studying in-domain predictive visual adaptation for multimodal emotion recognition, where the adapted visual representation is evaluated jointly with audio and text. We therefore use V-JEPA2-style predictive learning as an in-domain adaptation mechanism rather than introducing a new JEPA objective or claiming that predictive adaptation is universally preferable to other self-supervised objectives.

## 2.4 Adaptive Multimodal Fusion

Fusion remains a central problem in MER because each modality can be informative, ambiguous, or misleading depending on the sample. Classical approaches include early and late fusion, tensor-based fusion (Zadeh et al., 2017), attention-based fusion (Liang et al., 2018; Tsai et al., 2019), and gated fusion mechanisms (Arevalo et al., 2017). More recent methods emphasize reliability modeling through audio-guided fusion (Shi & Gao, 2024) and modality-specific dynamic emotion experts (Fang et al., 2025), moving beyond static fusion by adapting modality interactions for each example.

Emotion-JEPA follows this reliability-sensitive direction but uses audio as an explicit conditioning signal. AMHF combines audio-conditioned modulation, sparse cross-modal routing, memory-based refinement, learned confidence-normalized weighting, and progressive fusion to regulate how visual and textual evidence contribute to the final prediction. This design is motivated by the role of acoustic prosody as a useful cue in MER, while recognizing that modality usefulness can vary across samples and that audio conditioning is a design choice rather than a universal reliability assumption.

**Positioning:** Emotion-JEPA complements prior MER and LMM-based affective understanding work by studying affect-domain visual adaptation and audio-conditioned fusion under a controlled labeled-training protocol. It differs from pseudo-label-based systems by using unlabeled clips for representation adaptation rather than additional classification targets, and from generic fusion baselines by evaluating AMHF through component ablations, strict fusion controls, modulator variants, and routing diagnostics.

## 3 Methodology

**Overview:** Emotion-JEPA is a two-stage framework for multimodal emotion recognition. Stage 1 adapts a pretrained visual video encoder to the affective domain using unlabeled emotion-domain videos and a masked latent prediction objective. This stage is visual-only: it does not use audio, text, pseudo-labels, or manual emotion annotations. Stage 2 trains a supervised multimodal classifier that combines the adapted visual representation with pretrained audio and text representations through Audio-Modulated Hybrid Fusion (AMHF). AMHF consists of audio-conditioned grouped gating, sparse cross-modal expert routing, memory-based refinement, confidence-normalized interaction weighting, and residual progressive fusion. During inference, predictions are produced only from the fused classifier; auxiliary modality heads are used for training supervision and diagnostic analysis. The JEPA predictor and EMA target encoder are used only during Stage 1 and are discarded before supervised multimodal training. Figure 1 summarizes the overall pipeline.

### 3.1 Problem Setup and Notation

We formulate multimodal emotion recognition as supervised classification over $C = 6$ emotion categories in MER2024-SEMI (Lian et al., 2024b). The labeled dataset is

$$\mathcal{D} = \{(x_i^v, x_i^a, x_i^t, y_i)\}_{i=1}^N,$$

where $x_i^v$ denotes a video clip, $x_i^a$ its synchronized audio segment, $x_i^t$ the corresponding transcript tokens, and $y_i \in \{1, \ldots, C\}$ the emotion label. An unlabeled video set $\mathcal{U}_v = \{x_j^v\}_{j=1}^M$ is used only during Stage 1 for self-supervised visual adaptation.

Each modality $m \in \{v, a, t\}$ is processed by an encoder $f_m$ and projection head $P_m$:

$$h_m = P_m(f_m(x^m)), \qquad h_m \in \mathbb{R}^d, \quad d = 512. \tag{1}$$

All fusion operations in Stage 2 are performed in this shared embedding space.

Stage 1 adapts only the visual branch using $\mathcal{U}_v$. Stage 2 performs supervised multimodal learning on $\mathcal{D}$: projection heads, pooling modules, AMHF, auxiliary modality heads, and the fused classifier are trainable, while only the upper layers of the pretrained modality encoders are fine-tuned. Earlier encoder layers are frozen to preserve pretrained representations and reduce overfitting under limited labeled supervision. We restrict predictive adaptation to the visual encoder because the visual branch is especially affected by the gap between generic video pretraining and face-centric affect recognition, whereas the audio and text branches use large-scale pretrained speech and language encoders during supervised training.

The training and inference phases are separated in implementation: the JEPA predictor and EMA target encoder are used only during Stage 1, AMHF and supervised classifiers are trained only during Stage 2, and final inference uses fused classifier logits without auxiliary-head ensembling. A phase-wise summary of updated components and objectives is provided in Appendix H.1.

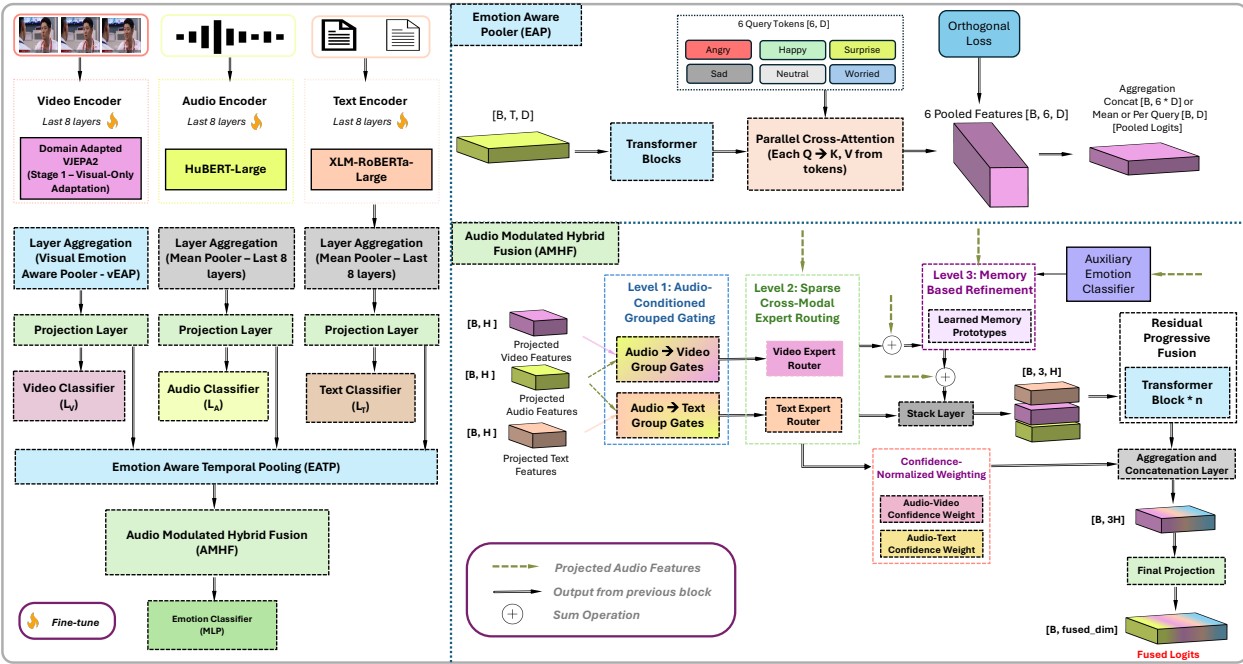

Figure 1: **Overview of Emotion-JEPA:** Stage 1 adapts a V-JEPA2-style visual encoder on unlabeled emotion-domain videos, and Stage 2 trains a supervised multimodal classifier with AMHF fusion. vEAP aggregates visual tokens, EATP pools temporally structured video/audio features, and text enters AMHF as an utterance-level feature. AMHF uses audio-conditioned grouped gating, sparse expert routing, memory-based refinement, confidence-normalized weighting, and residual fusion. Auxiliary heads support training and diagnostics; inference uses only the fused classifier.

### 3.2 Stage 1: Affective Visual Domain Adaptation

The goal of Stage 1 is to adapt the visual encoder to emotion-domain video before supervised multimodal training. Starting from a public V-JEPA2 ViT-G visual encoder, we use a V-JEPA2-style masked latent prediction objective as an in-domain adaptation mechanism rather than proposing a new JEPA objective. This stage is visual-only: it uses unlabeled emotion-domain videos without audio, text, manual emotion labels, or pseudo-labels. Since MER often depends on subtle facial movements, gaze changes, and low-intensity expression transitions, we continue training the visual encoder on unlabeled emotion-domain videos using masked latent prediction. Adaptation is performed on approximately 113k unlabeled clips from MER2024. Each clip is sampled at 4 FPS and resized or cropped to 256×256. We apply multi-scale spatiotemporal masking over a large portion of the video volume, encouraging the model to infer masked latent targets from visible temporal context.

Following the V-JEPA formulation (Bardes et al., 2024; Assran et al., 2025), each clip is divided into visible context regions and masked target regions. Let $f_\theta$ denote the student encoder, $f_\xi$ the exponential-moving-average target encoder, and $g_\theta$ the predictor. For each prediction mask $m \in \mathcal{M}$, let $z_m$ denote the predictor output for the masked target tokens, and let $\bar{h}_m$ denote the corresponding layer-normalized target-encoder representation after applying the same prediction mask. Stage 1 minimizes the latent MAE prediction loss

$$\mathcal{L}_{\text{pred}} = \frac{1}{|\mathcal{M}|} \sum_{m \in \mathcal{M}} \frac{1}{N_m d} \sum_{i=1}^{N_m} \left\| z_{m,i} - \text{sg}(\bar{h}_{m,i}) \right\|_1, \tag{2}$$

where $N_m$ is the number of masked target tokens for prediction mask $m$, $d$ is the latent dimension, and $\text{sg}(\cdot)$ stops gradients through the EMA target branch. The target representation $\bar{h}_m$ is obtained by layer-normalizing the target-encoder output and selecting the masked target tokens corresponding to the predictor output.

The target encoder parameters are updated by exponential moving average:

$$\theta_{\text{target}} \leftarrow \mu\theta_{\text{target}} + (1 - \mu)\theta_{\text{student}}, \qquad \mu = 0.9995. \tag{3}$$

During adaptation, only the upper transformer blocks of the visual encoder and the predictor are updated, while the lower visual layers remain frozen.

After adaptation, the resulting visual encoder $f_v^*$ is used in Stage 2 for supervised multimodal training. The predictor and EMA target encoder are discarded after Stage 1. Additional details on the trainable visual subset, masking policy, EMA update, optimization settings, and compute budget are provided in Appendix B.

### 3.3 Stage 2: Supervised Multimodal Learning

Stage 2 trains the multimodal emotion classifier using the labeled set $\mathcal{D}$. The adapted visual encoder $f_v^*$ processes the video stream, while HuBERT-Large and XLM-RoBERTa-Large provide pretrained audio and text representations. These encoders are used as strong unimodal feature extractors for the controlled fusion study, rather than being introduced as new representation models. Each modality is projected into the shared embedding space in Eq. 1, yielding $(h_v, h_a, h_t)$. During supervised training, the projection heads, emotion-aware pooling modules, AMHF, auxiliary modality heads, and fused classifier are trained, while only the upper layers of the pretrained modality encoders are fine-tuned. Video and audio are aligned at the clip level before fusion, while text is treated as an utterance-level semantic cue because the available transcripts do not provide reliable token-level timestamps. The resulting modality representations are pooled and projected into the shared embedding space before entering AMHF, as described next. Full details on modality encoders, projection heads, Stage 2 optimization, and training throughput are provided in Appendix C.

### 3.4 Emotion-Aware Pooling

Before fusion, sequence-level encoder outputs are converted into clip-level modality embeddings. We use emotion-aware pooling (EAP) as a learned query-based aggregation operator rather than relying only on average pooling. The queries are trainable parameters optimized from the MER objective; they are not ground-truth labels and are not conditioned on the test label during inference.

Given an input sequence $S \in \mathbb{R}^{L \times d_s}$ and learned emotion queries $Q \in \mathbb{R}^{C \times d_s}$, EAP computes

$$O = \text{MHA}(Q, S, S), \qquad \alpha = \text{softmax}(\psi(O)), \qquad \text{EAP}(S; Q) = \sum_{c=1}^{C} \alpha_c O_c, \tag{4}$$

where MHA denotes multi-head cross-attention, $\psi$ maps each query output to a scalar, and the softmax is taken over the $C$ query outputs. The resulting weighted sum produces a single clip-level representation.

EAP is used in two places. First, visual Emotion-Aware Pooling (vEAP) aggregates visual tokens from the adapted V-JEPA2 encoder before projection:

$$\bar{x}_v = \text{EAP}(X_v; Q_v), \qquad h_v = P_v(\bar{x}_v), \tag{5}$$

where $X_v = f_v^*(x^v)$ denotes the visual token sequence. Second, Emotion-Aware Temporal Pooling (EATP) denotes the application of the same query-based pooling operator after projection to temporally structured modality sequences before AMHF. In the final hybrid temporal configuration, EATP is applied only to projected video and audio sequences, which retain temporal structure. Text bypasses temporal alignment and EATP, entering AMHF as a projected utterance-level contextual feature; this avoids treating a global lexical embedding as a temporal sequence. For notational simplicity, we denote the resulting modality representations entering AMHF as $(h_v, h_a, h_t)$.

### 3.5 Audio-Modulated Hybrid Fusion

AMHF combines modality evidence through audio-conditioned, sample-adaptive fusion. It uses the audio representation as a conditioning signal to modulate and route visual and textual evidence, while assigning

learned confidence-normalized weights to the routed audio-video and audio-text interactions. As shown in Figure 1, AMHF consists of five components: audio-conditioned grouped gating, sparse cross-modal expert routing, memory-based refinement, confidence-normalized interaction weighting, and residual progressive fusion.

**Audio-conditioned grouped gating:** Given projected embeddings $h_v, h_a, h_t \in \mathbb{R}^d$, AMHF first uses the audio representation to predict group-level gates for the video and text representations. For each non-audio modality $x \in \{v, t\}$, $h_x$ is split into $G$ channel groups $\{h_x^{(g)}\}_{g=1}^G$. The audio embedding predicts group-level gates

$$c_x = \sigma(A_x h_a), \qquad A_x \in \mathbb{R}^{G \times d},$$

and the non-audio features are modulated as

$$\hat{h}_x^{(g)} = c_{x,g} h_x^{(g)}, \qquad x \in \{v, t\}, \quad g = 1, \ldots, G. \tag{6}$$

This allows acoustic evidence to condition visual and textual feature groups before cross-modal routing.

**Sparse cross-modal expert routing:** After grouped gating, AMHF routes each audio-modality interaction through a sparse mixture of experts. For each target modality $x \in \{v, t\}$, we form a paired interaction feature

$$p_x = [h_a \| \hat{h}_x] \in \mathbb{R}^{2d}.$$

The sparse router computes expert weights from the audio modulator stream, not from the concatenated pair:

$$\pi = \mathrm{softmax}(R h_a), \qquad R \in \mathbb{R}^{E \times d}.$$

Only the top-$k$ experts are selected, and each selected expert processes the paired audio-target interaction feature:

$$r_x = \sum_{j \in \mathrm{Top}\text{-}k(\pi)} \pi_j E_{x,j}(p_x), \qquad x \in \{v, t\}, \tag{7}$$

where each expert $E_{x,j} : \mathbb{R}^{2d} \to \mathbb{R}^d$ is an MLP. A lightweight load-balancing regularizer is used to reduce expert collapse. This sparse routing step uses the audio stream to select interaction experts, while the selected experts operate on the concatenated audio-video or audio-text features.

**Memory-based refinement:** AMHF uses a small set of learnable memory prototypes to provide a shared refinement space for recurring audio-conditioned interaction patterns. Let $M \in \mathbb{R}^{K \times d}$ denote $K$ learned memory slots. The projected audio representation attends to these slots,

$$\tilde{h}_a = \mathrm{MHA}(h_a, M, M), \qquad a_{\mathrm{mem}} = \mathrm{LN}(h_a + \tilde{h}_a), \tag{8}$$

where LN denotes layer normalization. The memory-refined audio representation is then stacked with the routed audio-video and audio-text interaction features,

$$S_{\mathrm{amhf}} = [a_{\mathrm{mem}}; r_v; r_t] \in \mathbb{R}^{3 \times d},$$

and passed to a lightweight transformer refinement block. The memory slots are learned model parameters used to refine the temporally pooled audio-conditioned interaction representation; they do not maintain recurrent state across frames, clips, or batches.

**Confidence-normalized interaction weighting:** AMHF assigns learned scalar weights to the routed audio–video and audio–text interaction features before final fusion. For each routed interaction, a lightweight MLP produces an attenuation score:

$$\rho_{av} = \sigma(\phi_{av}(r_v)), \qquad \rho_{at} = \sigma(\phi_{at}(r_t)), \tag{9}$$

where $\phi_{av}$ and $\phi_{at}$ are lightweight MLPs. These scores are converted into normalized interaction weights:

$$\omega_{av} = \frac{1 - \rho_{av}}{(1 - \rho_{av}) + (1 - \rho_{at}) + \epsilon}, \qquad \omega_{at} = \frac{1 - \rho_{at}}{(1 - \rho_{av}) + (1 - \rho_{at}) + \epsilon}. \tag{10}$$

The weights are learned fusion gates and should not be interpreted as calibrated probabilistic uncertainty estimates.

**Residual progressive fusion:** The final AMHF stage combines the interaction pathway with direct modality evidence. The confidence-weighted interaction representation is

$$z_{\text{int}} = P_f \left( [a_{\text{mem}} \| \omega_{av} r_v \| \omega_{at} r_t] \right), \tag{11}$$

where $P_f : \mathbb{R}^{3d} \to \mathbb{R}^d$. To preserve information that may not be captured by the routed interaction path, we also project the original modality embeddings:

$$z_{\text{res}} = P_{\text{res}}([h_v \| h_a \| h_t]), \qquad z_{\text{fused}} = z_{\text{int}} + \gamma z_{\text{res}}. \tag{12}$$

where $\gamma$ controls the strength of the residual pathway. In the submitted best configuration, we use $\gamma = 0.3$. The final embedding $z_{\text{fused}} \in \mathbb{R}^d$ is passed to a linear classifier for emotion prediction. AMHF hyperparameters, including the number of gate groups, experts, top-$k$, memory slots, residual fusion strength, and refinement layers, are listed in Appendix C.4.

### 3.6 Training Objective

Stage 2 is trained with a supervised fused classification loss, auxiliary modality-specific classification losses, and lightweight auxiliary regularization terms. The auxiliary modality heads encourage each branch to retain discriminative emotion evidence and support diagnostic analysis, but they are not used for inference and should not be interpreted as independently trained unimodal baselines.

The fused classifier predicts

$$\hat{y} = \text{softmax}(W_f z_{\text{fused}} + b_f), \tag{13}$$

and the compact Stage 2 objective is

$$\mathcal{L}_{\text{stage2}} = \mathcal{L}_{\text{fused}} + \lambda_{\text{mod}} \frac{\mathcal{L}_v + \mathcal{L}_a + \mathcal{L}_t}{3} + \lambda_{\text{aux}} \mathcal{L}_{\text{aux}}, \tag{14}$$

Here, $\mathcal{L}_{\text{fused}}$ is the supervised loss for the fused classifier, and $\mathcal{L}_v$, $\mathcal{L}_a$, and $\mathcal{L}_t$ are auxiliary modality-head losses. The auxiliary regularization term $\mathcal{L}_{\text{aux}}$ includes sparse-routing regularization and the active vEAP auxiliary orthogonality term. We use $\lambda_{\text{mod}} = 0.3$ and $\lambda_{\text{aux}} = 0.3$ in the submitted best configuration. Final inference uses only the fused classifier in Eq. 13; the implementation-level decomposition of $\mathcal{L}_{\text{aux}}$ is provided in Appendix C.5.

## 4 Experiments and Results

This section evaluates Emotion-JEPA on MER2024-SEMI and analyzes the contribution of its two central design choices: affect-domain V-JEPA2-style visual adaptation and structured audio-modulated multimodal fusion. We first report the main MER2024-SEMI comparison, then study fusion design through strict control variants, independent unimodal baselines, AMHF routing and modulator sensitivity, and visual adaptation diagnostics. Additional implementation details, extended comparisons, and auxiliary analyses are provided in the appendix.

### 4.1 Evaluation Setup

For evaluation, we use MER2024-SEMI six-class emotion recognition task (Lian et al., 2024b), where clips are labeled as Angry, Happy, Neutral, Sad, Surprise, or Worried. Following the benchmark protocol, the labeled set contains 5,030 clips, split into 4,275 training clips and 755 validation clips, with final evaluation on the 1,169-clip MER2024-SEMI test set. The benchmark also provides over 110k unlabeled clips, which Emotion-JEPA uses only for Stage 1 self-supervised visual adaptation; no pseudo-labels, self-training, voting, or ensembling are used during supervised Stage 2 training. We report weighted-average F1 (WAF) as the primary metric, together with accuracy and macro-F1 when available. Unless otherwise stated, controlled comparisons use the same labeled split, preprocessing pipeline, pretrained encoders, shared projection dimension, and evaluation protocol as the full Emotion-JEPA model; training and implementation details are provided in Appendix C.

## 4.2 Main Results on MER2024-SEMI

Table 1 situates Emotion-JEPA among representative MER2024-SEMI systems. The comparison includes methods that use the unlabeled pool in different ways, ranging from supervised benchmark baselines to pseudo-labeling, self-training, voting, and self-supervised representation adaptation. Emotion-JEPA uses the unlabeled clips only for Stage 1 visual adaptation and does not use pseudo-labels, self-training, voting, or ensembling during supervised Stage 2 training.

Table 1: Main comparison with representative MER2024-SEMI systems. WAF is the primary metric. The unlabeled-data column summarizes how each method uses the unlabeled pool.

| Method | Key technique | Unlabeled-data use | WAF (%, ↑) |
|---|---|---|---|
| MERTools Baseline (Lian et al., 2024b) | HuBERT + CLIP + cross-attention | none reported | 86.73 |
| EmoVCLIP (Qi et al., 2024) | V–L prompting + self-training | pseudo-labeling/self-training | 90.51 |
| Shi & Gao (Shi & Gao, 2024) | semi-supervised audio fusion | pseudo-labeling/semi-supervised learning | 89.83 |
| Ge et al. (Ge et al., 2024) | audio–text fusion + voting | voting/refinement over unlabeled data | 88.25 |
| Emotion-JEPA (ours) | V-JEPA2-style visual adaptation + AMHF | self-supervised visual adaptation only | 85.72 |

Emotion-JEPA obtains 85.72% WAF, 85.89% accuracy, and 84.25% macro-F1 using the submitted best checkpoint. While pseudo-labeling and voting-based systems achieve higher leaderboard scores, Emotion-JEPA remains competitive under a more controlled no-pseudo-labeling protocol. The result is close to the reported supervised MERTools baseline while using a different design centered on predictive visual adaptation and structured audio-modulated fusion. Therefore, we use Table 1 primarily for benchmark positioning, and rely on the controlled analyses below to examine how visual adaptation, fusion design, unimodal encoder strength, and routing choices contribute to the final performance.

## 4.3 Fusion Design Analysis

We next evaluate whether the AMHF fusion design contributes beyond the pretrained modality encoders and shared projection space. Table 2 compares the default audio-modulated AMHF configuration with simpler fusion alternatives and alternative AMHF modulator choices. These controls use the same MER2024-SEMI labeled split and the same modality encoders, while varying the fusion mechanism.

Table 2: Fusion design comparison on MER2024-SEMI. WAF is the primary metric.

| Fusion configuration | WAF (%, ↑) | Acc. (%, ↑) | Macro-F1 (%, ↑) |
|---|---|---|---|
| Concat-MLP late fusion | 80.31 | 81.01 | 78.45 |
| Cross-attention fusion | 80.56 | 80.84 | 79.21 |
| Simple audio-gated fusion | 81.92 | 82.55 | 80.09 |
| AMHF, video-modulated variant | 81.09 | 81.61 | 79.81 |
| AMHF, text-modulated variant | 82.97 | 83.58 | 79.84 |
| AMHF, audio-modulated default | **85.72** | **85.89** | **84.25** |

The results show that AMHF improves over concat-MLP late fusion, cross-attention fusion, and a simpler audio-gated fusion variant across WAF, accuracy, and macro-F1. The alternative modulator variants further indicate that the conditioning signal matters: using audio as the modulator performs better than using video or text in these runs. This supports the use of audio-conditioned fusion for MER2024-SEMI, where acoustic prosody provides strong affective information, while not implying that audio is universally the best conditioning signal across all MER settings.

## 4.4 Ablation Studies

Table 3 summarizes two complementary ablation analyses on MER2024-SEMI. Framework-level ablations evaluate pooling, face-centric preprocessing, Stage 1 visual adaptation, and encoder fine-tuning, while AMHF component ablations remove one fusion stage at a time. The results show that encoder fine-tuning and Stage 1

visual adaptation have large framework-level effects, while sparse routing, memory refinement, confidence-normalized weighting, and residual fusion all contribute to AMHF performance.

Table 3: Ablation studies on MER2024-SEMI. Δ WAF is measured relative to the corresponding full model in each block.

| Framework-level ablations | | | AMHF component ablations | | |
|---|---|---|---|---|---|
| Configuration | WAF (%, ↑) | Δ WAF | Configuration | WAF (%, ↑) | Δ WAF |
| Full Emotion-JEPA | **85.72** | – | Full AMHF | **85.72** | – |
| w/o vEAP | 85.45 | -0.27 | w/o grouped gating | 82.44 | -3.28 |
| w/o vEAP and EATP | 82.46 | -3.26 | w/o sparse routing | 80.16 | -5.56 |
| w/o face detection | 80.84 | -4.88 | w/o memory refinement | 80.39 | -5.33 |
| Public V-JEPA2, no Stage 1 | 77.80 | -7.92 | w/o confidence weighting | 80.39 | -5.33 |
| Frozen encoders | 72.07 | -13.65 | w/o residual fusion | 77.48 | -8.24 |

Overall, the ablations indicate that performance depends on both task-specific adaptation of the pretrained encoders and the combined AMHF fusion pathway, rather than on a single isolated component.

## 4.5 Independent Single-Modality Baselines and Auxiliary Diagnostics

To separate multimodal fusion effects from single-modality encoder strength, we evaluate independently trained audio-only and text-only baselines using the same MER2024-SEMI splits. We also report auxiliary modality heads inside Emotion-JEPA as diagnostic outputs. These auxiliary heads are trained jointly with the multimodal objective and are not used for final prediction; therefore, they should not be interpreted as standalone unimodal baselines.

Table 4: Unimodal evidence on MER2024-SEMI. Independent baselines are trained as standalone single-modality models. Auxiliary heads are diagnostic heads inside the jointly trained Emotion-JEPA model and are not used for final prediction.

| Model / head | Setting | Modality | WAF (%, ↑) | Acc. (%, ↑) |
|---|---|---|---|---|
| XLM-RoBERTa-Large text-only | independent | Text | 36.88 | 36.78 |
| HuBERT-Large audio-only | independent | Audio | 81.13 | 81.52 |
| Emotion-JEPA video auxiliary head | joint diagnostic | Video | 44.37 | 47.82 |
| Emotion-JEPA audio auxiliary head | joint diagnostic | Audio | 75.74 | 76.13 |
| Emotion-JEPA text auxiliary head | joint diagnostic | Text | 18.42 | 23.61 |
| Emotion-JEPA fused prediction | final model | V+A+T | **85.72** | **85.89** |

The independent baselines show that text alone is weak on MER2024-SEMI, while HuBERT-Large audio-only is strong, reaching 81.13% WAF. This confirms that the benchmark contains substantial acoustic affective signal and supports using audio as the AMHF conditioning stream. The auxiliary heads are weaker than independently optimized single-modality models because they are trained as regularizing and diagnostic heads inside a joint multimodal objective, not as standalone classifiers. The fused Emotion-JEPA prediction improves over both the audio auxiliary head and the independently trained HuBERT-only baseline, indicating that the final prediction benefits from multimodal integration rather than auxiliary-head ensembling.

**Corruption sensitivity diagnostics:** We also run test-time corruption diagnostics on the submitted Emotion-JEPA checkpoint by zeroing, noising, or shuffling individual modalities. Audio perturbations cause the largest WAF drops, visual perturbations cause moderate drops, and text perturbations have limited effect, consistent with the single-modality and auxiliary-head results in Table 4. These diagnostics are stress tests of modality dependence and should not be interpreted as calibration analysis of the confidence-normalized AMHF weights. Full perturbation results and internal-signal analyses are provided in Appendix F.

### 4.6 Sparse-Routing Sensitivity

We further analyze the sparse-routing design in AMHF by varying the number of experts and selected experts. Table 5 compares the default routing configuration with smaller, stricter, and larger routing variants. These experiments are sensitivity analyses rather than exhaustive hyperparameter searches.

Table 5: Sparse-routing sensitivity on MER2024-SEMI. The default row uses the canonical submitted best Emotion-JEPA checkpoint; the remaining rows are single-run routing variants.

| Configuration | Routing setting | WAF (%, ↑) |
|---|---|---|
| AMHF default | $E = 4$, top-$k = 2$ | **85.72** |
| AMHF, smaller routing | $E = 2$, top-$k = 1$ | 83.45 |
| AMHF, stricter sparsity | $E = 4$, top-$k = 1$ | 80.87 |
| AMHF, more experts | $E = 8$, top-$k = 2$ | 84.09 |

The default $E = 4$, top-$k = 2$ setting performs best among the evaluated routing variants. Selecting only one expert with $E = 4$ leads to a larger drop, while increasing to $E = 8$ does not improve performance, suggesting that the default setting provides a useful balance between capacity and selective interaction. As a separate backbone-control experiment, replacing the ViT-G visual backbone with a smaller ViT-L backbone while keeping AMHF fixed gives 82.12% WAF, indicating that visual backbone capacity also contributes to the final score.

### 4.7 Visual Adaptation Diagnostics

We next examine the role of Stage 1 visual adaptation. Table 6 reports frozen-probe diagnostics and full-pipeline comparisons. In the full multimodal pipeline, using the public V-JEPA2 backbone without Stage 1 adaptation gives 77.80% WAF, while the predictive-adapted Emotion-JEPA model reaches 85.72% WAF. As an additional objective-level control, we adapt the same V-JEPA2 visual backbone on the unlabeled MER2024 videos using a compute-bounded contrastive visual adaptation objective; implementation details are provided in Appendix B.7. This contrastive-adapted encoder reaches 78.28% WAF when used in the same downstream AMHF pipeline.

Table 6: Visual adaptation diagnostics on MER2024-SEMI. Frozen probes evaluate visual representations without supervised encoder fine-tuning, while full-pipeline results evaluate each visual representation inside the supervised AMHF pipeline.

| Visual representation | Evaluation | WAF (%, ↑) | Acc. (%, ↑) | Macro-F1 (%, ↑) |
|---|---|---|---|---|
| Public V-JEPA2 ViT-G | frozen probe | 56.08 | 59.37 | 48.05 |
| Predictive-adapted V-JEPA2 ViT-G | frozen probe | 56.29 | 59.11 | 48.87 |
| Contrastive-adapted V-JEPA2 ViT-G | frozen probe | 47.87 | 50.90 | 39.56 |
| Public V-JEPA2, no Stage 1 adaptation | full AMHF pipeline | 77.80 | 78.53 | 75.20 |
| Contrastive-adapted V-JEPA2 ViT-G | full AMHF pipeline | 78.28 | 78.61 | 75.71 |
| Predictive-adapted V-JEPA2 (ours) | full AMHF pipeline | **85.72** | **85.89** | **84.25** |

The frozen-probe results show only a small difference between the public and predictive-adapted V-JEPA2 representations, so we do not interpret frozen probing as standalone evidence that the adapted visual encoder is universally better. Instead, the full-pipeline comparison indicates that Stage 1 predictive adaptation is beneficial when used as part of the complete Emotion-JEPA adaptation-and-fusion pipeline. The contrastive adaptation result provides an additional objective-level control, and we treat this as a controlled but compute-limited diagnostic, rather than an exhaustive comparison of self-supervised adaptation objectives.

### 4.8 Model Complexity and Practical Cost

Emotion-JEPA uses large pretrained visual, audio, and text encoders, so the dominant computational cost comes from the backbone encoders rather than from AMHF alone. AMHF adds grouped gating, sparse expert

routing, memory-based refinement, confidence-normalized interaction weighting, and residual fusion in the shared embedding space. These modules increase the fusion-stage parameter count, but they are applied after modality projection and are small relative to the pretrained encoders. Stage 1 also requires a separate self-supervised adaptation pass over the unlabeled video pool; however, the JEPA predictor and EMA target encoder are discarded after adaptation and are not used during supervised Stage 2 training or inference. In practice, Emotion-JEPA should be viewed as a controlled adaptation-and-fusion framework rather than a parameter-efficiency method. Detailed trainable parameter counts, AMHF configuration, throughput, and implementation settings are provided in Appendix H.

**Additional analyses:** Additional comparisons with open-source LMM baselines, MER2025-SEMI transfer results, per-class metrics, confusion matrices, and full corruption-sensitivity diagnostics are provided in the appendix. We use these analyses for positioning and diagnostic interpretation rather than as the primary controlled evidence, because they involve different model families, transfer settings, or stress-test conditions.

### 4.9 Discussion

The results support three observations. First, Stage 1 visual adaptation is most useful in the complete multimodal pipeline: replacing the predictive-adapted visual encoder with the public V-JEPA2 backbone reduces WAF from 85.72% to 77.80%, while frozen-probe diagnostics show only a small standalone visual difference. Second, fusion design matters: AMHF improves over concat-MLP, cross-attention, and simpler audio-gated controls, and component ablations show contributions from sparse routing, memory-based refinement, confidence-normalized interaction weighting, and residual fusion. Third, MER2024-SEMI contains strong acoustic signal; the tuned HuBERT-only baseline reaches 81.13% WAF, but the full model remains higher, indicating benefit from multimodal integration beyond a strong audio encoder.

These findings should be interpreted within the controlled scope of the study. Emotion-JEPA uses unlabeled clips for self-supervised visual adaptation rather than pseudo-labeling, self-training, voting, or ensembling, and the contrastive adaptation baseline is compute-bounded rather than an exhaustive comparison of self-supervised objectives. The confidence-normalized AMHF weights are learned fusion gates, not calibrated uncertainty estimates. Corruption diagnostics show strong dependence on audio and moderate complementary value from visual information, but not uniform robustness to missing, noisy, or misleading modalities. The method also relies on large pretrained encoders, with approximately 1.9B total parameters and about 0.6B updated during Stage 2; cross-dataset generalization remains limited without target-domain fine-tuning, motivating future work on smaller encoders, cached features, distillation, pruning, and stronger transfer evaluation.

## 5 Conclusion and Future Work

This paper presented Emotion-JEPA, a two-stage framework for multimodal emotion recognition under limited labeled supervision. The framework first adapts a V-JEPA2-style visual encoder on unlabeled emotion-domain videos using masked latent prediction, then combines visual, acoustic, and textual representations through Audio-Modulated Hybrid Fusion. Experiments on MER2024-SEMI show that the full adaptation-and-fusion pipeline performs competitively under a no-pseudo-labeling protocol, with controlled analyses supporting the roles of Stage 1 visual adaptation, structured AMHF fusion, sparse routing, and multimodal integration beyond a strong audio-only baseline.

However, there are few limitations. The method relies on large pretrained encoders, face-centric preprocessing, and reasonably synchronized audio-video inputs. Its audio-conditioned design may be less effective when speech is missing, noisy, weakly affective, or temporally misaligned. More broadly, emotion labels are noisy task annotations and should not be interpreted as definitive measurements of a person's internal state; practical deployment should require informed consent, privacy-preserving data handling, demographic and cultural bias evaluation, and avoidance of high-stakes use as a final judgment of emotion. Future work should explore joint multimodal self-supervised adaptation, stronger missing-modality and misalignment robustness, demographic and cross-cultural evaluation, and transfer to fine-grained, open-vocabulary, descriptive, and cross-dataset emotion understanding benchmarks.

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

# Appendix

# A Dataset Description

## A.1 Labeled and Unlabeled Splits

We use the official MER2024 dataset. The labeled portion contains 5,030 clips, each with video, audio, and text transcripts. Following the benchmark protocol, we use 85% of the labeled set for training and 15% for validation, stratified by emotion class. Final evaluation is performed on the MER2024-SEMI test set with 1,169 labeled clips. The additional unlabeled set contains approximately 113k clips and is used only for Stage 1 self-supervised visual adaptation.

Each clip is approximately 4–6 seconds long and is annotated with one of six emotions: *Angry*, *Happy*, *Neutral*, *Sad*, *Surprise*, and *Worried*. We do not use pseudo-labeling, self-training, voting, or ensembling on the unlabeled or test sets.

Table 7 summarizes the class distribution for the 5,030-sample labeled set and the MER2024-SEMI test set.

Table 7: Class distribution for the MER2024 labeled set and MER2024-SEMI test set.

| Emotion | MER2024 Labeled (5,030) | | MER2024-SEMI Test (1,169) | |
|---|---|---|---|---|
| | Count | Pct. | Count | Pct. |
| Sad | 1,267 | 25.18% | 317 | 27.12% |
| Happy | 1,042 | 20.72% | 246 | 21.04% |
| Neutral | 901 | 17.92% | 201 | 17.19% |
| Angry | 703 | 13.98% | 163 | 13.94% |
| Worried | 670 | 13.32% | 159 | 13.60% |
| Surprise | 447 | 8.89% | 83 | 7.10% |
| **Total** | 5,030 | 100% | 1,169 | 100% |

## A.2 Per-Modality Preprocessing

**Video:** Clips are decoded at 4 FPS and resized to 256×256. We detect faces with MediaPipe using a confidence threshold of 0.5 and expand the bounding box by 40% to include head pose and shoulder context. If no face is detected, a centered crop is used. During training, we apply light augmentation, including random resized cropping with scale in $[0.7, 1.0]$, aspect-ratio jitter in $[0.9, 1.1]$, and horizontal flipping.

**Audio:** Audio is resampled to 16 kHz mono and truncated or padded to match the clip duration. We optionally apply cepstral mean and variance normalization (CMVN) to reduce channel and loudness variability in heterogeneous recordings.

**Text:** We use the official Chinese and English transcripts. Text is normalized and tokenized with XLM-RoBERTa-Large. We do not enforce forced alignment; text is treated as an utterance-level contextual cue and bypasses temporal alignment in Stage 2.

## A.3 Text Modality Characteristics and Limitations

Although MER2024 provides Chinese and English transcripts for all clips, the text modality has several dataset-specific limitations for fine-grained emotion recognition.

**Short and weakly emotional utterances:** As shown in Figure 2, most transcripts are short. The median length is 16 words for Chinese and 13 words for English in the training split, and 9 words for Chinese and 5 words for English in MER2024-SEMI. Many utterances contain only a few semantically neutral words, and spoken content is often task-oriented or contextually vague rather than emotionally expressive. This limits the discriminative power of the text encoder.

**Missing, partial, or weakly aligned transcriptions:** Some clips contain truncated text, transcription errors, missing segments, or ambiguous speaker boundaries. In addition, transcripts are not force-aligned with audio or video and often do not correspond to the moment of peak facial or vocal expression. For this reason, we treat text as a global contextual cue rather than a temporally aligned signal in Stage 2.

**Impact on fusion:** These properties help explain the weak text-only and text auxiliary-head results reported in Table 4. Text is therefore incorporated as a supporting modality, while AMHF can rely more strongly on acoustic and visual cues when they provide clearer affective evidence.

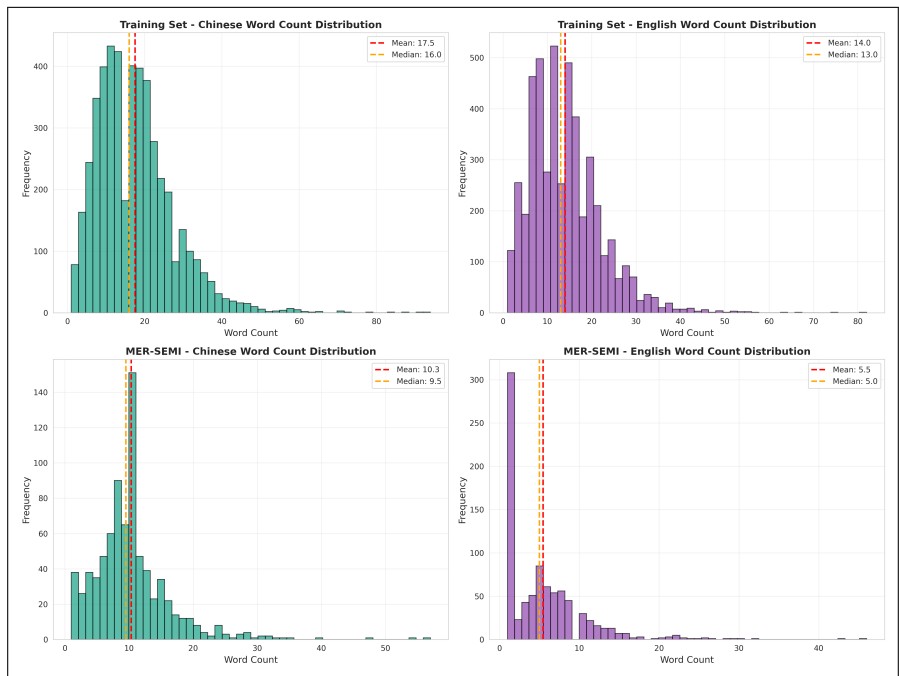

Figure 2: Word-count distributions for Chinese and English transcripts in the MER2024 training and MER2024-SEMI test splits. Most transcripts are short, which contributes to weak unimodal text discriminability.

# B    Additional Details for Stage 1: Predictive Visual Adaptation

## B.1    Model Overview and Trainable Subset

Stage 1 adapts the V-JEPA2 ViT-Giant visual backbone to the MER2024 video domain using masked latent prediction. In our configuration, the V-JEPA2 encoder contains 1,012,173,952 parameters ($\approx$1.01B), and the predictor network contains 22,381,312 parameters ($\approx$22M), for a total of approximately 1.03B parameters.

During visual adaptation, we partially unfreeze the encoder and train the predictor:

- the first 32 of 40 encoder layers remain frozen;
- the top 8 encoder layers are unfrozen;
- the predictor network is fully trainable;
- the EMA target encoder is updated without backpropagation.

Table 8 summarizes the parameter and compute breakdown for Stage 1. We use $\times$ to indicate non-trainable components and $\checkmark$ to indicate trainable ones.

Table 8: V-JEPA2 components active during Stage 1 visual adaptation. $\times$ indicates non-trainable components and $\checkmark$ indicates trainable ones. FLOPs are approximate forward-pass estimates.

| Component | Params | Trainable | % of Total | FLOPs/sample |
|---|---|---|---|---|
| Frozen encoder blocks (layers 1–32) | 810M | $\times$ | 78% | ~140 GF |
| Trainable encoder blocks (layers 33–40) | 202M | $\checkmark$ | 20% | ~35 GF |
| Predictor network | 22M | $\checkmark$ | 2% | ~10 GF |
| **Total (encoder + predictor)** | **1.03B** | – | **100%** | ~185 GF |
| **Trainable subset** | **224M** | – | **22%** | ~45 GF |

*Note:* FLOP values in Table 8 are approximate forward-pass estimates for individual components and are not used as training-time compute indicators. Practical cost is reported using throughput and GPU-hours in Appendix B.5.

## B.2 Masking Strategy and Sampling

We use a hierarchical masking policy with 14 spatiotemporal blocks per clip. The masks cover approximately 60–70% of the spatiotemporal volume and are tuned for short MER2024 clips sampled at 4 FPS. Table 9 summarizes the masking configuration.

Table 9: Multi-scale masking configuration used during Stage 1 visual adaptation.

| Block Type | # | Spatial | Temporal | Aspect | Purpose |
|---|---|---|---|---|---|
| Large temporal | 6 | 6–10% | 85–100% | 1.2–2.0 | Long-range motion |
| Medium temporal | 4 | 8–12% | 85–100% | 0.5–0.9 | Mid-range dynamics |
| Medium spatial | 3 | 18–25% | 70–100% | 0.8–1.2 | Structure + motion |
| Large spatial | 1 | 35–45% | 80–100% | 0.75–1.5 | Broad spatial context |
| **Total** | **14** | **60–70%** | **80–100%** | – | Multi-scale representation |

## B.3 EMA Target Network

We use the standard JEPA student–teacher setup, where a momentum-updated target encoder provides the latent prediction targets:

$$\theta_{\text{tgt}} \leftarrow m\theta_{\text{tgt}} + (1 - m)\theta_{\text{stu}}, \qquad m = 0.9995.$$

Only the student encoder and predictor participate in backpropagation. The EMA target encoder provides stable latent targets during visual adaptation and is discarded after Stage 1.

## B.4 Training Configuration

Stage 1 uses the following settings:

- Batch size: 24 per GPU, 96 effective across 4 GPUs.

- Prediction loss: latent MAE/L1 loss between predictor outputs and layer-normalized masked target-encoder representations.

- Optimizer: AdamW with weight decay 0.04.

- Learning rate: $2 \times 10^{-4}$ with cosine decay and 2-epoch warmup.

- Precision: BF16.

- Augmentations: random resized cropping with scale in $[0.7, 1.0]$, aspect-ratio jitter in $[0.9, 1.1]$, and horizontal flipping.

- Duration: 6 visual-adaptation epochs with iterations per epoch (IPE) set to 800.

## B.5 Compute and Throughput

Stage 1 visual adaptation was conducted on 4×A100 40GB GPUs. Since only the top 8 encoder blocks and the predictor are trainable, this stage is lighter than full-network visual pretraining. All measurements are taken from the steady-state training phase.

- Throughput: approximately 95 clips/s.

- Peak memory: approximately 28 GB per GPU.

- Total compute: approximately 164 GPU-hours for 6 adaptation epochs with IPE of 800.

The FLOP estimates in Table 8 are forward-pass estimates only. Practical training cost is better reflected by throughput and GPU-hours.

## B.6 Impact on Downstream Performance

The downstream effect of Stage 1 is evaluated in the main paper through the framework-level ablation and visual adaptation diagnostics. In the full AMHF pipeline, using the predictive-adapted visual encoder gives 85.72% WAF, while replacing it with the public V-JEPA2 backbone without Stage 1 adaptation gives 77.80% WAF. This corresponds to a +7.92 WAF difference in the full multimodal setting. As discussed in Section 4.7, this result should be interpreted together with the frozen-probe and contrastive-adaptation diagnostics rather than as a universal claim about all visual self-supervised objectives.

## B.7 Contrastive Visual Adaptation Control

To provide an objective-level control for Stage 1, we also adapt the same public V-JEPA2 ViT-G visual backbone on the unlabeled MER2024 video pool using a contrastive visual adaptation objective. This control uses the same unlabeled data source, face-centric preprocessing, backbone family, and upper-block update policy as the predictive Stage 1 setting. It is intended as a compute-bounded diagnostic control rather than an exhaustive or fully compute-matched comparison of self-supervised video objectives.

For each unlabeled video clip, two stochastic augmented views are sampled. Each view is encoded by the visual backbone, mean-pooled over visual tokens, and passed through a two-layer projection head. Let $q_i^{(1)}$ and $q_i^{(2)}$ denote the $\ell_2$-normalized projected representations of the two views of clip $i$. We optimize a symmetric NT-Xent loss:

$$\mathcal{L}_{\text{con}} = -\frac{1}{2N} \sum_{i=1}^{N} \left[ \log \frac{\exp(\text{sim}(q_i^{(1)}, q_i^{(2)})/\tau)}{\sum_{j=1}^{N} \exp(\text{sim}(q_i^{(1)}, q_j^{(2)})/\tau)} + \log \frac{\exp(\text{sim}(q_i^{(2)}, q_i^{(1)})/\tau)}{\sum_{j=1}^{N} \exp(\text{sim}(q_i^{(2)}, q_j^{(1)})/\tau)} \right], \tag{15}$$

where $N$ is the global batch size after distributed all-gather, $\tau = 0.2$, and $\text{sim}(\cdot, \cdot)$ denotes cosine similarity. The projection head is used only during contrastive adaptation and is discarded before downstream MER training. After adaptation, the contrastive-adapted visual encoder is inserted into the same downstream AMHF pipeline used by the predictive-adapted encoder.

# C Additional Details for Stage 2: Multimodal Training

## C.1 Evaluation Protocol and Implementation Details

All MER2024-SEMI results are evaluated on the official six-class emotion recognition setting. The six emotion categories are angry, happy, neutral, sad, surprise, and worried. We report weighted-average F1 (WAF) as the primary metric, following the benchmark convention, and also report accuracy and macro-F1

when available. WAF is emphasized because the labeled set is class-imbalanced, while macro-F1 provides an additional view of performance across classes independent of class frequency.

Unless otherwise stated, all ablation experiments use the same preprocessing pipeline, data split, modality encoders, projection dimension, and evaluation protocol as the full Emotion-JEPA model. The shared embedding dimension is set to $d = 512$. The visual branch uses the predictive-adapted V-JEPA2 encoder from Stage 1, while the audio and text branches use HuBERT-Large and XLM-RoBERTa-Large, respectively. For supervised Stage 2 training, the projection heads, emotion-aware pooling modules, AMHF fusion module, auxiliary modality heads, and fused classifier are trainable. Only the upper layers of the pretrained modality encoders are fine-tuned, while earlier layers remain frozen.

The fused classifier is used for final prediction at inference time. Auxiliary modality heads are included during supervised training to encourage modality-specific discriminative features and to support diagnostic analysis, but they are not used as an inference ensemble and should not be interpreted as independently trained unimodal baselines. Therefore, all reported main results correspond to the fused classifier output unless explicitly stated otherwise.

Class weighting is used during supervised training to reduce the effect of label imbalance. Additional regularization and optimization settings follow the submitted best configuration, including the auxiliary modality-loss weight $\lambda_{\mathrm{mod}} = 0.3$. Routing-related regularization is applied only when the corresponding AMHF component is active. The same evaluation script and metric computation are used for the full model and ablation variants to ensure direct comparability.

### C.2 Modality Encoders

**Video:** We use the predictive-adapted V-JEPA2 ViT-Giant encoder from Stage 1. Face-centric crops are resized to $256 \times 256$ and fed as 16-frame clips. During Stage 2 supervised training, the upper visual Transformer blocks are fine-tuned, while the JEPA predictor and EMA target encoder used in Stage 1 are not used for multimodal classification or inference.

**Audio:** HuBERT-Large (Hsu et al., 2021) processes 16 kHz waveforms with a 25 ms window and 10 ms hop. We extract hidden states from the last eight Transformer layers and apply mean pooling across time to obtain a 1024-D representation.

**Text:** XLM-RoBERTa-Large (Conneau et al., 2020) encodes Chinese and English transcripts. We average hidden states from the final eight layers and mean-pool across tokens, yielding a 1024-D textual embedding. Text is treated as an utterance-level contextual feature rather than a temporally aligned sequence.

### C.3 Projection and Shared Embedding Space

Each modality output is projected into a shared $d{=}512$-dimensional space:

$$\mathrm{Proj}(x) = \mathrm{GELU}\big(\mathrm{LN}(Wx)\big),$$

where $W$ is a learnable linear mapping and LN is LayerNorm. The projected features $(\mathbf{v}, \mathbf{a}, \mathbf{t}) \in \mathbb{R}^d$ are used by modality-specific auxiliary classifiers and as inputs to the Audio-Modulated Hybrid Fusion (AMHF) module.

### C.4 AMHF Hyperparameters

AMHF operates in the shared embedding space with the following configuration:

- **Audio-conditioned grouped gates:** $G{=}8$ groups, each in $\mathbb{R}^{64}$.

- **Sparse routing experts:** $E{=}4$ experts with hidden size $h{=}512$; top-$k{=}2$ experts are selected per sample.

- **Memory-based refinement:** $K{=}16$ learned memory slots in the submitted best configuration.

- **Refinement Transformer:** 3 encoder layers, 8 heads, hidden size $h{=}512$.

- **Residual fusion strength:** $\gamma = 0.3$.

The memory slots are learned prototype-like parameters used for refinement of temporally pooled audio-conditioned interaction features. They are not recurrent temporal states, external memory entries, class labels, or additional supervision.

## C.5 Stage 2 Objective Details

For completeness, the auxiliary regularization term in Eq. 14 is decomposed as

$$\mathcal{L}_{\text{aux}} = \mathcal{L}_{\text{route}} + \lambda_{\text{orth}}\mathcal{L}_{\text{vEAP-orth}}, \tag{16}$$

where $\mathcal{L}_{\text{route}}$ is the sparse-routing load-balancing regularizer and $\mathcal{L}_{\text{vEAP-orth}}$ denotes the auxiliary orthogonality penalty returned by the visual emotion-aware pooler. We use $\lambda_{\text{orth}} = 0.5$ in the submitted best configuration.

The vEAP orthogonality term contains query- and output-diversity components. The query-diversity scalar is computed without gradient flow to the query tokens, while the output-diversity component is differentiable and contributes through the auxiliary-loss path. Temporal EATP orthogonality is disabled in the submitted best configuration. No contrastive loss or explicit uncertainty-calibration loss is used during Stage 2.

## C.6 Training Configuration

Stage 2 uses the following settings:

- **Batch size:** 12 per GPU with $3\times$ gradient accumulation, giving an effective batch size of 144 across 4 GPUs.

- **Optimizer:** AdamW with weight decay 0.05.

- **Learning rate:** $2 \times 10^{-4}$ with warmup-cosine decay, 3 warmup epochs, and minimum LR $1 \times 10^{-6}$.

- **Layer-wise LR decay:** factor 0.95 applied to all encoders.

- **Precision:** BF16.

- **Regularization:** dropout 0.15, stochastic depth 0.2, label smoothing 0.1, and video-stream Mixup/CutMix with $\alpha_{\text{mix}} = 0.2$ and $\alpha_{\text{cut}} = 0.3$.

- **Training duration:** 80 epochs with early stopping; the submitted best checkpoint is selected at epoch 14.

**Video-stream Mixup and CutMix:** During Stage 2 training, Mixup and CutMix are applied only to the video stream before the multimodal forward pass. At each training step, Mixup is sampled with probability 0.5 when enabled; if Mixup is not selected, CutMix is sampled with probability 0.5. For Mixup, two video clips $x_i^v$ and $x_j^v$ are linearly combined as

$$\tilde{x}_i^v = \lambda x_i^v + (1 - \lambda)x_j^v, \qquad \lambda \sim \text{Beta}(\alpha_{\text{mix}}, \alpha_{\text{mix}}). \tag{17}$$

For CutMix, a spatial crop is replaced across the video tensor using another clip from the batch, and $\lambda$ is adjusted according to the retained visible area. Audio and text are not mixed and remain the original paired streams for sample $i$. The fused and auxiliary classification losses use the same mixed-label criterion,

$$\mathcal{L}_{\text{mix}} = \lambda\ell(\hat{y}_i, y_i) + (1 - \lambda)\ell(\hat{y}_i, y_j), \tag{18}$$

where $\ell$ denotes the Stage 2 classification loss.

### C.7 Compute and Throughput

Stage 2 multimodal training was performed on the same 4×A100 40 GB workstation used in Stage 1; hardware details are provided in Appendix H. Because only the upper visual blocks, selected HuBERT and XLM-RoBERTa-Large layers, and the supervised fusion modules are trainable, Stage 2 is lighter than full-network multimodal training but heavier than feature-only fusion training.

- **Throughput:** approximately 32–35 clips/s aggregate.

- **Peak memory:** approximately 28–30 GB per GPU.

- **Training duration:** approximately 9–11 GPU-hours for a full run, with the best checkpoint selected after approximately 2.5 hours.

Given the partially frozen encoders and multimodal architecture, FLOP counts are not reported. Throughput and GPU-hours provide a more reliable measure of practical compute.

### C.8 Training Dynamics

Figure 3 presents the Stage 2 training curves, including WAF, accuracy, macro-F1, precision-recall, loss trajectories, and the learning-rate schedule. Validation metrics rise steadily during the first 10–15 epochs, and no divergence or unstable optimization behavior is observed.

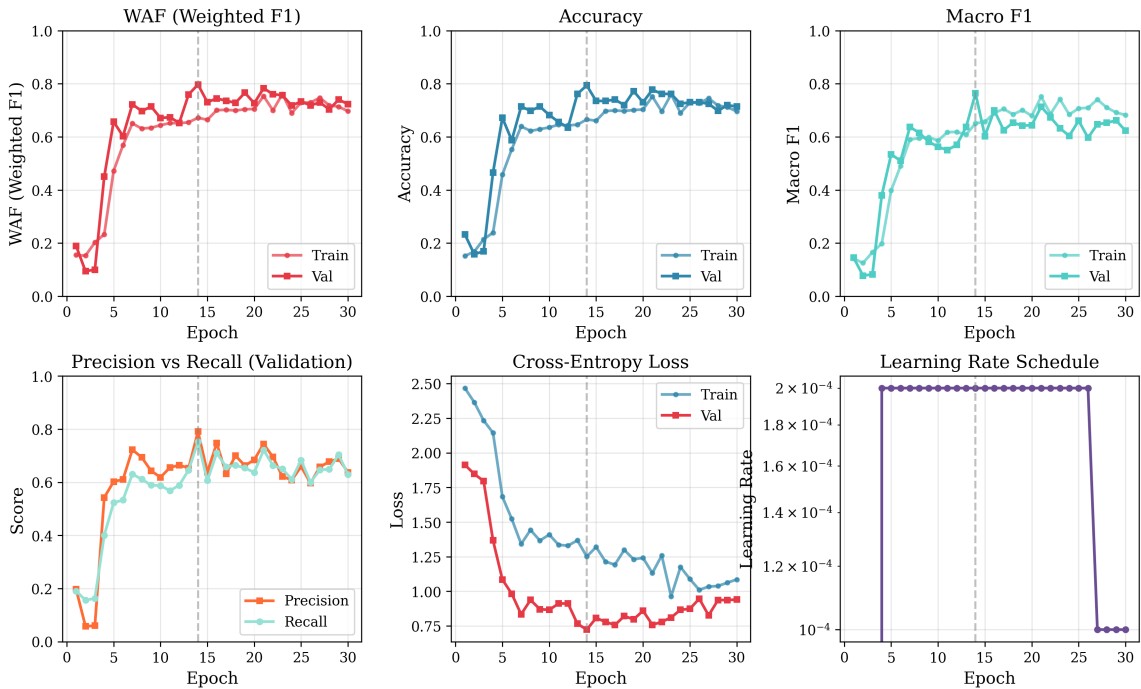

Figure 3: Extended Stage 2 training curves showing WAF, accuracy, macro-F1, precision-recall, loss, and learning-rate schedule.

## D Large Multimodal Model Evaluation Details

We include LMM comparisons as an additional positioning analysis rather than as primary controlled evidence. The evaluated LMMs differ from Emotion-JEPA in model family, pretraining data, modality support, prompting format, and fine-tuning protocol. Therefore, these results should be interpreted as context for current multimodal foundation models on MER2024-SEMI, not as capacity-matched baselines.

All evaluated LMM baselines, including Apollo, Qwen-VL, and Qwen-Omni, accept raw video input directly. Qwen-Omni additionally supports raw audio input. We therefore do not apply manual windowing, frame segmentation, or model-specific sampling heuristics; each model processes the full clip as provided, subject to its native preprocessing pipeline.

### D.1 Prompt Templates

To ensure a consistent evaluation format across LMMs, we use a unified modality-aware instruction template. The A+V+T, A+T, and V+T variants share the same output format and differ only in the sensory cues made available to the model.

**Unified template (abridged):**

> *"You are an expert in emotion recognition. Based on the provided modality input, classify the dominant emotional state as one of: **happy, sad, angry, worried, surprised, neutral**. Respond with one word."*

**Modality-specific cues:**

- **Audio-Video-Text (A,V,T):** the model is instructed to consider facial expressions, body language, eye behavior, gestures, tone of voice, and transcript content.

- **Audio-Text (A,T):** visual cues are removed, while prosodic cues and transcript content are retained.

- **Video-Text (V,T):** audio cues are removed, while facial, gesture, body-language, and transcript cues are retained.

**Full verbatim prompts:** Full verbatim prompt strings used for LMM inference under different modality settings are provided as plain-text files in the supplementary archive under `appendix/prompts/`.

### D.2 LMM Results on MER2024-SEMI

Table 10 reports the LMM results used for appendix-level positioning. Zero-shot LMMs perform substantially below the task-specific Emotion-JEPA model. LoRA fine-tuning improves Qwen-Omni, especially in audio-supported settings, but the best LoRA-tuned LMM remains below Emotion-JEPA on WAF and macro-F1. These results suggest that task-specific adaptation and structured fusion remain useful for MER2024-SEMI, although the comparison is not controlled for architecture, pretraining data, or modality support.

Table 10: Large multimodal model results on MER2024-SEMI. These results are included for positioning and are not capacity-matched controlled baselines.

| Model | Setting | Modality | Acc. (%, ↑) | WAF (%, ↑) | Macro-F1 (%, ↑) |
|---|---|---|---|---|---|
| Apollo 3B | zero-shot | V+T | 51.92 | 51.92 | 44.84 |
| Apollo 7B | zero-shot | V+T | 46.54 | 45.47 | 38.19 |
| Qwen-VL 3B | zero-shot | V+T | 55.21 | 55.06 | 49.57 |
| Qwen-VL 7B | zero-shot | V+T | 55.38 | 55.53 | 49.33 |
| Qwen-Omni 7B | LoRA fine-tuned | A+T | 79.21 | 79.16 | 77.63 |
| Qwen-Omni 7B | LoRA fine-tuned | A+V+T | 78.87 | 78.56 | 73.80 |
| Emotion-JEPA | supervised task-specific | A+V+T | **85.89** | **85.72** | **84.25** |

### D.3  LoRA Fine-Tuning Configuration for Qwen-Omni

We fine-tune Qwen2.5-Omni 3B and 7B with parameter-efficient LoRA adapters using the `LLaMA-Factory` framework. We use the MER2024 labeled training split with 5,030 clips formatted in the multimodal `ShareGPT` conversation style. Each training example contains a user instruction requesting an emotion prediction with `<video>` and `<audio>` tags, a single-word assistant response containing the ground-truth emotion, and top-level fields specifying the corresponding video and audio paths.

**Adapter parameters:**

- Rank $r$: 16.

- Scaling factor $\alpha$: 16.

- Target modules: attention and MLP projections, expanded from `"all"` to `q_proj`, `k_proj`, `v_proj`, `o_proj`, `mlp.gate_proj`, `mlp.up_proj`, and `mlp.down_proj`.

**Training hyperparameters:**

- Learning rate: $1.0 \times 10^{-5}$ with AdamW, cosine schedule, and warmup ratio 0.1.

- Precision: FP16.

- Per-device batch size: 1 with gradient accumulation of 4, giving an effective batch size of 8 on 2 GPUs.

- Epochs: 2.0, corresponding to approximately 2-3 effective passes depending on the internal train/validation split.

**Hardware:** Fine-tuning runs are executed on up to 2×NVIDIA A100 40 GB GPUs, with typical VRAM usage in the 20-40 GB range depending on model size and checkpointing configuration.

## E  MER2025-SEMI Transfer Results

We also evaluate Emotion-JEPA on MER2025-SEMI as an appendix-level transfer check. This analysis is included to assess whether the MER2024-trained framework transfers to a newer benchmark setting, but it is not used as the primary controlled evidence because the dataset, split, and benchmark protocol differ from MER2024-SEMI.

Table 11 reports three settings: the official MER2025 baseline, direct transfer of the MER2024-trained Emotion-JEPA model, and Emotion-JEPA after supervised fine-tuning on MER2025. Direct transfer performs below the official baseline, indicating a dataset shift between MER2024-SEMI and MER2025-SEMI. After supervised fine-tuning, Emotion-JEPA slightly exceeds the official MER2025 baseline, suggesting that the adaptation-and-fusion design remains useful when retuned to the target benchmark.

Table 11: MER2025-SEMI transfer results. These results are included as an appendix-level transfer check rather than as primary MER2024-SEMI evidence.

| Method / setting | WAF (%, ↑) | Acc. (%, ↑) |
|---|---|---|
| Official MER2025 baseline | 78.63 | 78.77 |
| Emotion-JEPA, direct transfer | 74.47 | 74.98 |
| Emotion-JEPA, supervised fine-tuning | **79.15** | **79.07** |

These results indicate that Emotion-JEPA does not transfer perfectly without target-domain supervision, but supervised fine-tuning recovers and slightly improves performance relative to the official MER2025 baseline. We therefore interpret this experiment as evidence of transfer feasibility after target-domain adaptation, not as a claim of strong zero-shot cross-benchmark generalization.

# F Corruption Sensitivity and Internal-Signal Diagnostics

## F.1 Perturbation Protocol

We evaluate modality dependence using controlled test-time perturbations applied to the submitted full-model checkpoint. The clean setting uses the original video, audio, and transcript inputs. For missing-modality tests, *audio zero* replaces the waveform with zeros, *video zero* replaces the visual input with zeros, and *text blank* replaces the transcript with an empty string. For degraded or misleading-modality tests, *audio noise* replaces the waveform with Gaussian noise with standard deviation 0.05, while *audio shuffle*, *video shuffle*, and *text shuffle* replace the corresponding modality with content from another sample. Shuffling preserves realistic modality content but breaks cross-modal and label correspondence.

## F.2 Extended Perturbation Results

Table 12 reports the full perturbation results, including combined missing-modality settings. These diagnostics characterize modality dependence under controlled corruption and should not be interpreted as evidence of calibrated uncertainty or uniform robustness. In the table, *zero* replaces the input with zeros, *noise* replaces audio with Gaussian noise, *shuffle* replaces a modality with content from another test sample, and *blank* replaces the transcript with an empty string. Combined settings apply the corresponding missing-modality operations simultaneously.

Table 12: Controlled modality perturbations on MER2024-SEMI using the submitted full-model checkpoint. Δ WAF is measured relative to the clean setting.

| Input condition | WAF (%, ↑) | Δ WAF |
|---|---|---|
| Clean | **85.72** | – |
| Audio zero | 27.99 | -57.73 |
| Audio noise | 37.29 | -48.43 |
| Audio shuffle | 41.26 | -44.46 |
| Video zero | 69.07 | -16.65 |
| Video shuffle | 72.23 | -13.49 |
| Text blank | 83.65 | -2.07 |
| Text shuffle | 83.52 | -2.19 |
| Audio zero + video zero | 0.94 | -84.78 |
| Audio zero + text blank | 26.84 | -58.88 |
| Video zero + text blank | 70.39 | -15.32 |

The results show that audio perturbations cause the largest degradation, visual perturbations cause moderate degradation, and text perturbations have limited effect. This pattern is consistent with the independent unimodal and auxiliary-head diagnostics in the main paper: MER2024-SEMI contains strong acoustic affective signal, while text is a weaker standalone cue. The combined missing-modality settings further show that severe removal of multiple modalities can lead to substantial failure, especially when audio is removed together with video.

## F.3 Internal-Signal Analysis

We also analyze AMHF internal signals, including sparse-routing weights, confidence-normalized interaction weights, gating statistics, and cross-modal interaction summaries. For each perturbation setting, we evaluate associations between these signals and fused or modality-head correctness using Spearman correlation, bootstrap confidence intervals, permutation tests, and false-discovery-rate correction. We report and interpret only statistically supported associations after correction.

This analysis is used as a post-hoc diagnostic to examine whether AMHF internal signals vary systematically under modality corruption. It is not used as an additional training objective, and the confidence-normalized interaction weights should not be interpreted as calibrated probabilistic uncertainty estimates.

# G  Additional Results

## G.1  Per-Class Metrics

Figure 4 reports per-class precision, recall, and F1 scores for the submitted Emotion-JEPA checkpoint on MER2024-SEMI. These results provide a class-level view of performance beyond the aggregate WAF, accuracy, and macro-F1 scores reported in the main paper.

Figure 4: Per-class precision, recall, and F1 scores for Emotion-JEPA on MER2024-SEMI.

## G.2  Oracle Top-$K$ Analysis

In addition to standard Top-1 evaluation, we report oracle Top-$K$ metrics for the single-label MER2024-SEMI task. A prediction is counted as correct if the ground-truth class appears among the model's Top-$K$ probability-ranked outputs. This does not change the task into multi-label evaluation; it measures how often the correct label remains among the model's highest-ranked alternatives.

Table 13 summarizes oracle Top-$K$ WAF and accuracy. Moving from Top-1 to Top-2 improves WAF from 85.72% to 95.52%, and Top-3 reaches 97.93%. This suggests that many remaining errors occur among closely ranked alternatives rather than being uniformly distributed across unrelated classes.

Table 13: Oracle Top-$K$ metrics on MER2024-SEMI. Top-1 corresponds to standard single-label evaluation.

| Metric | Top-1 | Top-2 | Top-3 |
|---|---|---|---|
| WAF (%, ↑) | 85.72 | 95.52 | 97.93 |
| Accuracy (%, ↑) | 85.89 | 95.22 | 97.65 |

Figure 5 visualizes the per-class Top-$K$ trends. The largest Top-$K$ gains appear for classes with higher ambiguity in the Top-1 predictions, such as *Neutral*, *Worried*, and *Angry*.

## G.3  Qualitative Success and Failure Cases

Figure 6 presents representative success and failure cases from MER2024-SEMI. Each example includes sampled video frames, the corresponding waveform and spectrogram, the transcript when available, and the model's probability distribution over the six emotion classes.

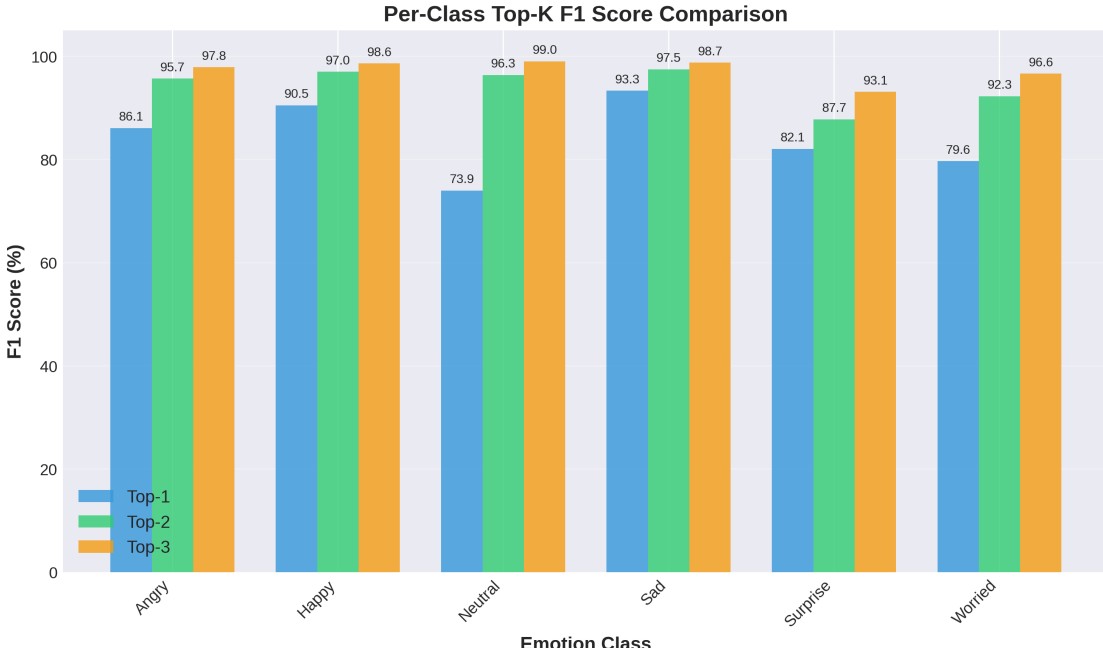

Figure 5: Per-class Top-$K$ F1 scores on MER2024-SEMI.

**Successful predictions:**  The model performs well when visual affect, prosody, and transcript cues are mutually supportive. For example, `samplenew3_00104570` (*Happy*) and `samplenew3_00032314` (*Sad*) illustrate cases where facial expressions and acoustic patterns provide consistent affective evidence, even when the transcript is limited.

**Failure cases:**  Errors often occur in ambiguous or low-arousal regions where facial motion, prosody, and conversational context provide weak or conflicting evidence. In `samplenew3_00097648`, the model predicts *Worried* despite largely neutral visual cues. In `samplenew3_00026688`, the model predicts *Sad*, possibly reflecting low lighting and subdued facial motion that make the neutral label difficult to distinguish.

**Interpretation:**  Across failure cases, the ground-truth label often remains among the higher-ranked alternatives, consistent with the oracle Top-$K$ analysis in Appendix G.2. These examples suggest that some errors reflect intrinsic ambiguity among closely related or low-arousal emotion categories. They also motivate future work on conflict-aware fusion and more explicit modeling of uncertain or ambiguous affective labels.

### G.4  Additional Confusion Matrices

**Ablation confusion matrices:**  To visualize how major components affect inter-class confusions, Figure 7 (7a,7b, 7c) shows normalized confusion matrices for three representative ablations: removing Stage 1 visual adaptation, replacing AMHF with cross-attention fusion, and freezing all encoders.

**LMM baselines vs. Emotion-JEPA:**  Figure 7d (7e, 7f, 7c) summarizes normalized confusion matrices for representative ablations, LMM baselines, and Emotion-JEPA. The top row shows how major ablations affect inter-class confusions, while the bottom row compares Qwen-Omni 7B zero-shot and LoRA-tuned baselines with the final Emotion-JEPA model.

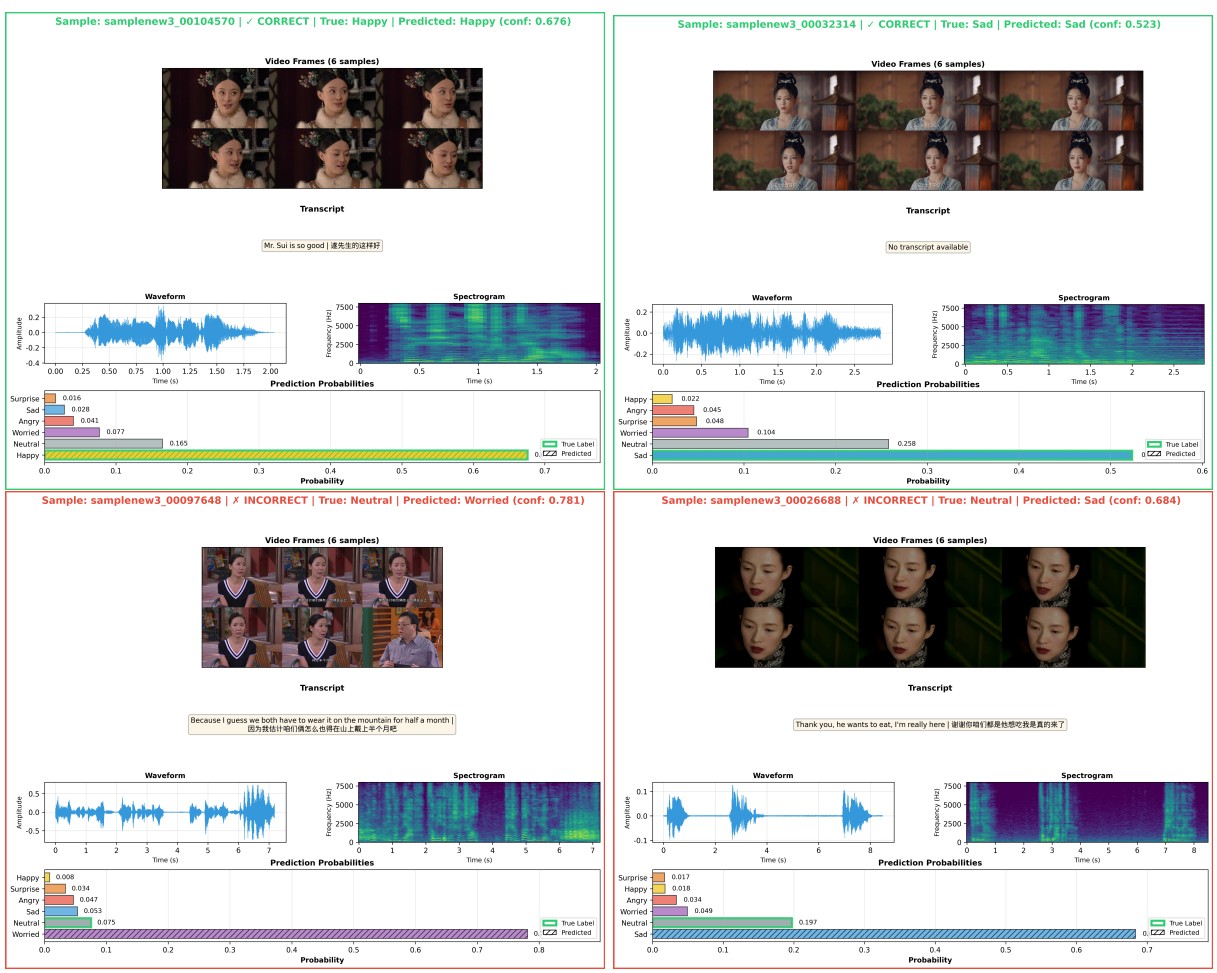

Figure 6: Qualitative examples from MER2024-SEMI. Top: success cases where Emotion-JEPA correctly predicts the target emotion. Bottom: failure cases illustrating ambiguity and possible modality conflict.

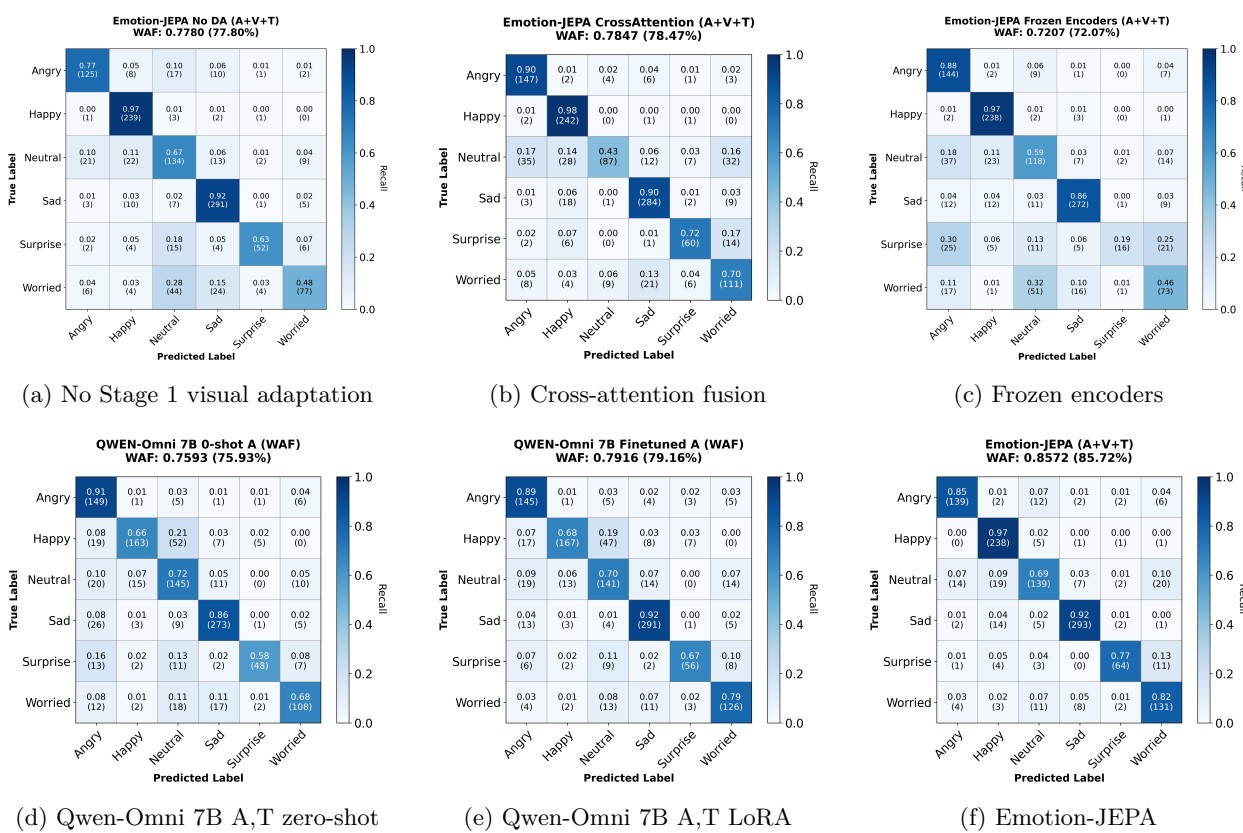

(a) No Stage 1 visual adaptation  (b) Cross-attention fusion  (c) Frozen encoders

(d) Qwen-Omni 7B A,T zero-shot  (e) Qwen-Omni 7B A,T LoRA  (f) Emotion-JEPA

Figure 7: Normalized confusion matrices on MER2024-SEMI. Top row: representative ablation settings showing the effects of removing Stage 1 visual adaptation, replacing AMHF with cross-attention fusion, and freezing all encoders. Bottom row: Qwen-Omni 7B zero-shot and LoRA-tuned LMM baselines compared with Emotion-JEPA.

# H Compute and Reproducibility

## H.1 Training and Inference Phases

Table 14 summarizes which components are updated in each phase of Emotion-JEPA. This separation clarifies that the JEPA predictor and EMA target encoder are used only during Stage 1 visual adaptation, while AMHF is trained during supervised Stage 2 multimodal learning and used at inference.

Table 14: Training and inference phases of Emotion-JEPA.

| Phase | Data | Updated components | Objective / output |
|-------|------|--------------------|--------------------|
| Stage 1 | Unlabeled videos | Upper visual encoder blocks and JEPA predictor; EMA target updated without gradients | Masked latent prediction |
| Stage 2 | Labeled video, audio, text | Upper encoder layers, projection heads, EAP modules, AMHF, auxiliary heads, and fused classifier | Supervised fused and auxiliary classification losses |
| Inference | Test video, audio, text | None | Fused classifier logits |

## H.2 Parameter and Compute Summary

Table 15 summarizes the practical parameter and compute profile of Emotion-JEPA. The model uses large pretrained visual, audio, and text encoders, so the dominant parameter count comes from the backbone encoders rather than from AMHF alone.

Table 15: Approximate parameter and compute summary for the submitted Emotion-JEPA configuration.

| Setting | Active parameters | Updated parameters | Practical compute |
|---------|-------------------|--------------------|-------------------|
| Stage 1 visual adaptation | ≈1.03B | ≈224M | ≈164 GPU-hours |
| Stage 2 multimodal training | ≈1.9B | ≈0.6B | ≈9–11 GPU-hours |
| Inference | ≈1.9B | 0 | single forward pass |

## H.3 Hardware and Runtime

Emotion-JEPA experiments were conducted on a workstation equipped with 4×NVIDIA A100-SXM4 40 GB GPUs. The software environment was:

- **OS:** Ubuntu 22.04.5 LTS.

- **GPU driver:** 550.144.03.

- **CUDA:** 12.4 system installation; 11.8 PyTorch runtime.

- **Frameworks:** Python 3.12, PyTorch 2.1, cuDNN 8.7.

Approximate runtimes are:

- **Stage 1 visual adaptation:** approximately 164 GPU-hours for 6 epochs with 24 clips/GPU, giving an effective batch size of 96.

- **Stage 2 multimodal training:** approximately 9–11 GPU-hours on 4×A100 GPUs, with the best checkpoint selected after approximately 2.5 hours via early stopping.

Peak GPU memory usage during Stage 2 was approximately 28–32 GB per GPU. Throughput and memory measurements were collected using `nvidia-smi` and the PyTorch profiler.

### H.4  Determinism and Seeding

We fix the global random seed to 42 for all Emotion-JEPA experiments. Determinism is enabled where supported:

- `torch.use_deterministic_algorithms(True)`.

- cuDNN deterministic mode enabled and benchmark mode disabled.

- Reproducible dataloader seeds using `worker_seed = base_seed + worker_id`.

Some low-level CUDA kernels, such as atomic operations inside attention modules, may remain nondeterministic. For this reason, rerun controls are interpreted as controlled sensitivity analyses rather than exhaustive statistical estimates.

### H.5  Implementation Notes

Emotion-JEPA builds on the official V-JEPA2 codebase and standard PyTorch-based libraries, including `timm`, `transformers`, `librosa`, and `mediapipe` for face detection and preprocessing. LoRA fine-tuning of Qwen2.5-Omni models is performed with the `LLaMA-Factory` toolkit. Hyperparameters for optimizer settings, learning-rate schedules, masking policies, partial-freezing rules, AMHF configuration, and LoRA fine-tuning are provided in the main paper and appendix to support reproducible implementation and evaluation.

