# OpenReview forum: "Emotion-JEPA: Predictive Visual Adaptation and Audio-Modulated Fusion for Multimodal Emotion Recognition"
_TMLR — Under review for TMLR_

### Review · Reviewer_1cw8 · 2026-06-16

**Summary Of Contributions:**

This paper proposes a two-stage framework for multimodal emotion recognition called Emotion-JEPA. Its primary contribution lies in utilizing masked latent prediction techniques to process unlabeled video segments, adapting a V-JEPA-style visual encoder to the emotion domain video, aiming to better capture subtle facial dynamics without pseudo-labels or additional manual annotation. Secondly, this paper proposes an audio modulation hybrid fusion module that uses audio as a reliability signal, modulating the fusion of visual and textual information through mechanisms such as gating, routing, temporal memory, uncertainty estimation, and progressive fusion. Furthermore, this paper emphasizes controlled empirical evaluation, distinguishing the impact of visual domain adaptation from the impact of the fusion mechanism through ablation experiments and baseline matching comparisons.

A key strength of this paper is its systematic study of representation adaptation and fusion design, rather than simply scaling the model. Ablation experiments show that both predictive visual adaptation and the proposed audio-guided fusion module significantly contribute to performance. This paper also pays special attention to differentiating its method from pseudo-label-based semi-supervised methods, which helps clarify the type of supervision used in practice.

Potential limitations include the complexity of the proposed fusion module, which comprises many interacting components that may be difficult to interpret or reproduce. Furthermore, the method relies on a large pre-trained encoder, face-centric preprocessing, and relatively synchronized multimodal inputs, which could limit its robustness in poorly controlled environments. Finally, while the results are competitive, the method does not surpass the most advanced pseudo-label-based MER2024 system; therefore, its primary value lies more in being a controlled study of adaptive and reliability-aware fusion than a definitive state-of-the-art achievement.

**Audience:**

Yes

**Audience Explanation:**

Some readers of TMLR may find the findings of this paper interesting, as it explores two broadly relevant problems in multimodal learning: domain adaptation of pre-trained representations and reliability-aware fusion across modalities. This work is particularly relevant to readers interested in multimodal emotion recognition, self-supervised video representation learning, and controlled evaluation of fusion architectures. Its key findings—that intra-domain predictive visual adaptation and audio conditional fusion can achieve significant performance improvements under limited labeled supervision—may have applications beyond this specific benchmark setting.

Compared to general machine learning papers, this paper is more specialized, and its impact will likely be primarily felt in areas such as emotion computing, multimodal representation learning, or robust multimodal fusion. Nevertheless, the controlled comparisons between predictive adaptation, cross-attention fusion, and the proposed AMHF module make the findings appealing to a significant portion of the TMLR readership.

**Broader Impact Concerns:**

I believe there are no major, broader implications that require in-depth discussion; a brief overview of the limitations of this paper will suffice. This paper primarily focuses on the technical aspects of multimodal emotion recognition, visual adaptation, and fusion design.

Nevertheless, given that emotion recognition may involve sensitive information in practice, the authors might briefly mention some common concerns such as privacy, informed consent, demographic or cultural biases, and misuse in high-risk decision-making. In my view, it is sufficient to briefly state that the model's predictions should not be interpreted as a final judgment of an individual's internal emotional state.

**Claims And Evidence:**

Yes

**Claims Explanation:**

The main arguments are generally supported by clear ablation experiments and control experiments. This paper provides direct evidence that predictive visual adaptation improves performance, with a 7.92 improvement in WAF compared to the domain-free adaptation variant; furthermore, AMHF also improves the WAF by 7.25 compared to the capacity-matched cross-attention fusion baseline. The paper also includes component-level ablation experiments showing that gating, routing, temporal memory, uncertainty estimation, and progressive fusion all contribute to the final results. These experiments lend considerable persuasiveness to the core arguments.

However, the evidence is not entirely conclusive. The approach is complex, and some claims regarding “reliability-aware” behavior are primarily based on performance ablation experiments rather than deeper diagnostic evidence, such as whether the model can correctly estimate modal reliability. Furthermore, the paper's performance does not surpass the most powerful pseudo-label-based MER2024 system; therefore, its claims should be considered support for the advantages of controlled adaptation/fusion, rather than a benchmark for state-of-the-art technology.

**Requested Changes:**

Please clarify the innovations of this paper and its position relative to previous work. This paper combines V-JEPA-style visual adaptation, a pre-trained audio/text encoder, emotion-aware pooling, and an audio-guided fusion module; however, the current presentation fails to clearly articulate which parts are new, which are inherited from previous work, and which design choices are primarily for engineering integration. The authors should more clearly distinguish their contributions from existing MER systems, audio-guided or reliability-aware fusion methods, and recent predictive video models for facial expression or emotion recognition. A clearer positioning will help readers understand whether the main innovations lie in the prediction adaptation phase, the AMHF design, the controlled evaluation scheme, or a combination of these elements.

Please strengthen the empirical support for the "reliability-aware" interpretation of AMHF. This is crucial to my recommendation. Current ablation experiments show that AMHF improves the WAF value compared to cross-attention mechanisms, and its components contribute to performance. However, these results do not adequately demonstrate that the learned gating, routing decisions, or uncertainty estimates correspond to meaningful modal reliability. The authors should supplement the diagnostic analysis to demonstrate AMHF's performance in situations where a particular modality is noisy, missing, insufficiently informative, or misleading. For example, they could report performance under corrupted audio/video/text input, analyze whether uncertainty scores are correlated with errors in a specific modality, or provide qualitative examples of the model correctly reducing the weights of unreliable visual or textual cues. This would make the claims regarding reliability perception more convincing.

Improving the reproducibility and clarity of the method description would enhance this work. AMHF comprises several interacting components, including audio spectrum gating, adaptive routing, temporal memory, uncertainty estimation, and progressive fusion. The entire system consists of two training phases, with the encoder partially frozen. While many implementation details have been provided, the method would be more complete if clearer pseudocode, a more explicit training/inference flow, and a precise description of the parameters updated at each phase were provided. The authors should also clarify the roles of the auxiliary classifier, regularization loss, sentiment-aware pooling, and the difference between inference and training time. This would make the paper more reproducible and reduce ambiguity regarding how the results are obtained.

Furthermore, the authors should better discuss model complexity, computational cost, and practical trade-offs. This will enhance the persuasiveness of this paper. The proposed system uses a large pre-trained encoder and a relatively complex fusion module, therefore it is necessary to clarify whether the performance gains are sufficient to offset the increased complexity. The authors should provide more direct comparisons with simpler alternatives, such as late-stage fusion, gated fusion, audio-guided attention mechanisms that do not require routing or memory, and smaller encoder configurations. They should also discuss the relationship between the number of parameters, trainable parameters, GPU cost, and inference time overhead and the observed performance gains. This will help readers assess the value of Emotion-JEPA primarily as a controlled science research project, a practical MER model, or both.

---

> ### Author Response · Authors · 2026-07-22
> **Response to Reviewer 1cw8**
>
> We thank the reviewer for the careful and constructive assessment. We appreciate the reviewer’s recognition that the paper is best understood as a controlled study of affect-domain visual adaptation and audio-modulated fusion, rather than as a leaderboard-oriented pseudo-labeling system. In the revision, we clarified the contribution, added diagnostic analyses, improved reproducibility details, and expanded the discussion of complexity and limitations. We address the main concerns point by point below.
>
> ### 1. Innovation and positioning
>
> We revised the Introduction and Related Work to more clearly distinguish inherited components from the contributions of this work. The revised manuscript now states that the JEPA objective and student–target training scheme are inherited from V-JEPA/V-JEPA2, while our contribution is to study V-JEPA2-style in-domain visual adaptation for MER and combine it with structured audio-modulated fusion under a controlled no-pseudo-labeling protocol. We also clarify that HuBERT-Large and XLM-RoBERTa-Large are used as pretrained audio/text encoders for a controlled fusion study, not introduced as new encoders.
>
> **Where addressed:** Introduction; Sections 2.2–2.4; positioning paragraph at the end of Section 2; Section 4.2 and Table 1.
>
> ### 2. Empirical support for AMHF behavior
>
> We strengthened the AMHF analysis in two ways. First, we added controlled fusion comparisons against concat-MLP late fusion, cross-attention fusion, and a simpler audio-gated fusion variant, as well as video- and text-modulated AMHF variants. In these controls, audio-modulated AMHF obtains 85.72% WAF, compared with 80.31% for concat-MLP, 80.56% for cross-attention, 81.92% for simple audio-gated fusion, 81.09% for video-modulated AMHF, and 82.97% for text-modulated AMHF. Second, we added corruption sensitivity diagnostics by zeroing, noising, or shuffling modalities. These diagnostics show that audio perturbations cause the largest degradation, visual perturbations cause moderate degradation, and text perturbations have limited effect.
>
> We also revised the terminology to soften our claim. The AMHF weights are now described as learned confidence-normalized interaction weights or fusion gates, not calibrated uncertainty estimates.
>
> **Where addressed:** Section 3.5; Section 4.3 and Table 2; Section 4.5; Appendix F.
>
> ### 3. Reproducibility and method clarity
>
> We revised the Methodology and Appendix to make the two-stage training and inference flow more explicit. The paper now clarifies that Stage 1 uses only unlabeled videos, updates the upper visual encoder blocks and predictor, and discards the predictor and EMA target encoder before Stage 2. Stage 2 trains the supervised multimodal classifier with AMHF, auxiliary heads, and the fused classifier, while inference uses only the fused classifier output.
>
> We also clarified the role of auxiliary heads, the decomposition of the auxiliary objective, and the data flow around EATP. In the submitted configuration, EATP is applied only to video and audio, while text bypasses EATP and enters AMHF as an utterance-level contextual feature. We further corrected implementation-level details: the Stage 1 loss is now described as latent MAE/L1 prediction against layer-normalized masked target features, and the AMHF memory size is reported using the submitted checkpoint configuration.
>
> **Where addressed:** Sections 3.2, 3.4, 3.5, and 3.6; Appendix B; Appendix C; Appendix H.1.
>
> ### 4. Complexity and practical trade-offs
>
> We added a dedicated complexity discussion and an appendix compute summary. The revised paper reports the approximate active and updated parameter counts, Stage 1 and Stage 2 GPU-hour costs, and practical throughput. We also added simpler fusion controls and a smaller ViT-L backbone control to help contextualize the performance–complexity trade-off. The revised manuscript explicitly states that Emotion-JEPA should be viewed primarily as a controlled adaptation-and-fusion framework rather than a lightweight deployment model.
>
> **Where addressed:** Section 4.3 and Table 2; Section 4.6 and Table 5; Section 4.8; Appendix H.
>
> ### 5. Broader-impact limitations
>
> We added a broader limitation statement noting that emotion labels are noisy task annotations and should not be interpreted as definitive measurements of a person’s internal state. The conclusion now also states that deployment should require informed consent, privacy-preserving data handling, demographic and cultural bias evaluation, and avoidance of high-stakes use as a final judgment of emotion.
>
> **Where addressed:** Section 5.
>
> Overall, the revised manuscript presents Emotion-JEPA with narrower and more precise claims: it studies affect-domain V-JEPA2-style visual adaptation and structured audio-modulated fusion for MER under limited labeled supervision, without claiming universal predictive-learning superiority, calibrated uncertainty estimation, or state-of-the-art leaderboard performance.

---

### Review · Reviewer_RZAH · 2026-07-09

**Summary Of Contributions:**

The paper studies multimodal emotion recognition under limited supervision. It proposes a two-stage framework: first, it performs in-domain self-supervised adaptation of the V-JEPA2 visual encoder on unlabeled emotion-domain videos using masked latent prediction; then, it trains a supervised multimodal classifier based on Audio-Modulated Hybrid Fusion (AMHF). The paper mainly demonstrates the feasibility of V-JEPA2-style predictive visual adaptation for affective video representation learning and introduces an audio-conditioned multimodal fusion method. The experimental results show that both visual adaptation and AMHF improve performance in the ablation studies, but the method does not outperform representative MER2024-SEMI systems and is even below the MERTools baseline, which also does not use pseudo-labeling.

**Additional Comments:**

1. The training objective includes $L_{orth}$ and $L_{bal}$, but both losses are incompletely defined.
2. The data flow around EATP is unclear. The main text states that temporal pooling is applied only to video and audio, but Figure 1 places EATP below all three projected modality branches, which makes it appear that text also enters EATP.
3. $\sigma$ is not explicitly defined.

**Audience:**

No

**Audience Explanation:**

The paper is currently insufficient to generate substantial interest among TMLR readers. Although it targets multimodal emotion recognition, self-supervised video adaptation, and multimodal fusion, its main contributions are relatively incremental. In addition, the overall experimental results do not show a clear advantage, as the method underperforms representative systems. The paper mainly relies on internal ablation studies to show that its components are useful, but these results only demonstrate that the components contribute within the proposed system and do not sufficiently establish broader empirical value.

**Claims And Evidence:**

No

**Claims Explanation:**

1. The text auxiliary head is not an independently trained text-only model, but an auxiliary head trained jointly with the fused classifier. Although the paper demonstrates through experiments and data analysis that text may be suppressed by the main fusion target during joint training, this does not represent the true upper limit of the text.

2. The paper emphasizes that modality reliability varies across samples, but AMHF fixes audio as the anchor. If the audio signal is noisy, weakly expressive, or temporally misaligned with the video, this assumption may not hold.

3. The paper lacks necessary architectural comparisons. It does not sufficiently demonstrate the advantage of AMHF over alternatives such as text-conditioned or vision-conditioned.

4. The adaptive routing module uses (E=4) MLP experts and selects top-(k=2) experts, but the paper does not justify this configuration. It also provides no evidence that the experts learn distinct multimodal interaction patterns.

5. The distinction between pseudo-labeling and JEPA-style adaptation is valid: the former converts unlabeled samples into supervised classification data, while the latter performs self-supervised representation learning. However, both use a large amount of unlabeled data. The paper should not use “no pseudo-labeling” to downplay this point, especially since its performance is lower than the MERTools baseline, which also does not use pseudo-labeling. Proving the feasibility of JPEA alone is not enough to demonstrate contribution 1.

6. The MER2025 experiment is insufficient to demonstrate generalization. MER2024-SEMI and MER2025-SEMI have similar task settings, the direct transfer result drops substantially, and the fine-tuned result is better understood as target-domain adaptation with only a small performance gain.

**Requested Changes:**

**Critical for Acceptance**
1. An independently trained text-only baseline should be added, rather than only reporting the text auxiliary head from joint training.
Additional architectural comparisons are needed to demonstrate that the audio-conditioned design is superior to other conditioned fusion variants.
2. Sensitivity analysis over (E) and top-(k) should be added. The paper should also report expert utilization, expert selection distribution, or specialization analysis to show that different experts exhibit distinct multimodal interaction patterns.
3. The distinction between pseudo-labeling and JEPA-style adaptation should be presented more fairly. Both methods use unlabeled data; they differ mainly in the form of supervision signal. The authors should explain why JEPA-style adaptation still constitutes a sufficient contribution despite its lower performance.

**Would Strengthen the Work**
1. More rigorous cross-dataset generalization experiments should be added.

---

> ### Author Response · Authors · 2026-07-22
> **Response to Reviewer RZAH**
>
> We thank the reviewer for the detailed feedback. The revision directly addresses the concerns about unimodal evidence, audio conditioning, fusion comparisons, routing sensitivity, pseudo-labeling language, and generalization. We also narrowed claims where the evidence did not support the stronger interpretation in the original submission.
>
> ### 1. Independent text-only baseline and auxiliary heads
>
> We added independently trained single-modality baselines and clearly separate them from auxiliary heads in the jointly trained model. The revised manuscript reports an independent text-only baseline at 36.88% WAF and 36.78% accuracy, and an independent audio-only baseline at 81.13% WAF. Auxiliary heads are now described only as training and diagnostic heads, not as independently optimized unimodal models or inference ensembles.
>
> **Where addressed:** Section 4.5 and Table 4; Section 3.6; Appendix C.1.
>
> ### 2. Audio as the AMHF conditioning stream
>
> We revised the paper to present audio conditioning as a design choice for MER2024-SEMI, not a universal reliability assumption. The strong independent audio-only baseline supports audio as a useful conditioning stream for this benchmark, while the corruption diagnostics show that performance degrades when audio is missing, noisy, or mismatched. We now discuss audio dependence as a limitation rather than implying that audio is always the most reliable modality.
>
> **Where addressed:** Section 4.5; Appendix F; Sections 4.9 and 5.
>
> ### 3. Architectural comparisons
>
> We added the requested fusion controls: concat-MLP late fusion, cross-attention fusion, simple audio-gated fusion, video-modulated AMHF, and text-modulated AMHF. The canonical audio-modulated AMHF model obtains 85.72% WAF, compared with 80.31%, 80.56%, 81.92%, 81.09%, and 82.97% WAF for the alternatives. These comparisons are now described as controlled reruns with shared encoders, split, projection dimension, and evaluation protocol, rather than as a fully capacity-matched proof.
>
> **Where addressed:** Section 4.3 and Table 2; Section 4.8; Appendix H.
>
> ### 4. Routing sensitivity and expert interpretation
>
> We added sparse-routing sensitivity over the number of experts and top-k selected experts. The default \(E=4\), top-\(k=2\) setting performs best among the tested settings, while stricter sparsity or more experts reduce WAF. We also narrowed the interpretation: routing is now presented as a sparse interaction-capacity mechanism, not as definitive evidence that experts learn human-interpretable specialized patterns.
>
> **Where addressed:** Section 4.6 and Table 5; Section 4.9.
>
> ### 5. Pseudo-labeling versus JEPA-style adaptation
>
> We revised the text to state the distinction more fairly. Emotion-JEPA uses unlabeled MER2024 clips for self-supervised visual representation adaptation rather than converting them into supervised pseudo-labels, but we agree that both approaches use unlabeled data. The paper now avoids using "no pseudo-labeling" to downplay this point and positions Emotion-JEPA below stronger pseudo-labeling/voting systems and the MERTools no-pseudo baseline when appropriate. The contribution claim is now limited to evaluating V-JEPA2-style in-domain adaptation as part of the full adaptation-and-fusion pipeline.
>
> **Where addressed:** Introduction; Section 2.1; Section 4.2 and Table 1; Section 4.7 and Table 6.
>
> ### 6. Generalization and method clarity
>
> We moved MER2025-SEMI from the main results to the appendix and present it as a transfer check. Direct transfer drops below the MER2025 baseline, while supervised fine-tuning recovers performance; we therefore treat cross-dataset generalization as a limitation rather than a core claim. We also clarified the Stage 1 latent MAE/L1 loss, Stage 2 auxiliary objective, routing regularization, vEAP orthogonality, and EATP flow. In the submitted hybrid temporal configuration, EATP is applied to video/audio, while text enters AMHF as an utterance-level contextual cue.
>
> **Where addressed:** Sections 3.2-3.6; Section 4.9; Appendix B; Appendix C.4-C.5; Appendix E; Figure 1 caption.
>
> Overall, the revised manuscript presents a more modest and better-supported claim: Emotion-JEPA is a controlled study of in-domain visual adaptation and structured audio-modulated fusion, not a claim of universal audio reliability, calibrated uncertainty, state-of-the-art leaderboard performance, or strong zero-shot generalization.

---

### Review · Reviewer_wpyQ · 2026-07-11

**Summary Of Contributions:**

This paper proposes Emotion-JEPA, a two-stage framework for multimodal emotion recognition. The first stage adapts a V-JEPA2 visual encoder on unlabeled MER2024 videos using masked latent prediction. The second stage combines visual, audio, and text representations using the proposed Audio-Modulated Hybrid Fusion module, which includes audio-conditioned gating, expert routing, memory, uncertainty weighting, and progressive fusion. The experiments include framework and component ablations, comparisons with large multimodal models, and transfer evaluation on MER2025-SEMI.

**Audience:**

Yes

**Audience Explanation:**

Multimodal emotion recognition under limited labeled supervision is relevant to researchers working on multimodal representation learning, self-supervised video adaptation, affective computing, and adaptive fusion. In particular, investigating whether unlabeled in-domain video adaptation can improve multimodal emotion recognition without pseudo-labeling is a useful research question. The design space of asymmetric, audio-conditioned fusion may also be of interest because modality quality can vary considerably across emotion-recognition examples.
Therefore, I believe that at least a subset of the TMLR audience would be interested in the findings. My negative assessment is based on the gap between the paper’s claims and its current evidence, rather than on a lack of potential audience interest or on the fact that the method does not establish a new state of the art.

**Broader Impact Concerns:**

The submission does not currently provide an adequate discussion of the ethical and societal implications of multimodal emotion recognition.

**Claims And Evidence:**

No

**Claims Explanation:**

1. The Stage-1 experiment does not isolate the effect of the predictive JEPA objective. The principal comparison is between a model that receives substantial additional in-domain training on approximately 113k clips and a model without this adaptation. Therefore, the reported 7.92 WAF difference supports the narrower conclusion that this particular in-domain continued-training procedure is useful, but it does not establish that latent prediction is the important factor. There is no compute-, data-, and architecture-matched comparison against other adaptation objectives, such as masked reconstruction, contrastive learning, teacher-student feature matching, or a simpler continued-pretraining control. Moreover, the paper infers that the adapted encoder learns more “affect-sensitive visual representations” from the performance of the complete multimodal system, without a controlled visual-only linear-probe or fine-tuning analysis before and after adaptation.
2. The evidence does not validate the characterization of AMHF as reliability-aware or uncertainty-aware. The quantities called uncertainties in Eqs. 7-8 are sigmoid outputs of learned MLPs, but no uncertainty supervision, probabilistic derivation, calibration objective, or calibration evaluation is provided. It is therefore unclear whether these values measure uncertainty rather than functioning as unconstrained attention gates.
3. The claimed capacity-matched comparison with cross-attention is not sufficiently documented. The paper does not report the exact trainable parameter counts, depth, width, number of attention blocks, computational cost, or hyperparameter-selection procedure for AMHF and the cross-attention baseline. Given that AMHF includes multiple MLP experts, memory parameters, Transformer refinement layers, gating networks, uncertainty networks, and residual projections, this information is necessary for interpreting the reported 7.25 WAF difference.
4. Modality-contribution analysis is not conclusive. The reported unimodal results are auxiliary heads jointly trained with the fused system, rather than independently optimized unimodal models. They therefore cannot establish the actual capability of each individual modality or the dependence of the complete model on that modality. Independently trained unimodal models, leave-one-modality-out evaluations, and controlled modality corruptions would provide much stronger evidence.

Overall, the current results show that the authors’ complete implementation performs competitively and that removing several components reduces performance in a single experimental run. They do not yet provide convincing evidence for the stronger mechanistic conclusions concerning predictive learning, affect-specific representations, calibrated uncertainty, temporal stabilization, or reliability-aware routing.

**Requested Changes:**

1. Isolate the contribution of predictive visual adaptation. Add data-, architecture-, and compute-matched adaptation baselines using at least one reasonable alternative self-supervised objective. A visual-only evaluation before and after adaptation, using both a frozen linear probe and a controlled fine-tuning protocol, is also needed. If these experiments cannot be provided, the claims should be narrowed from the benefits of “predictive adaptation” and “affect-sensitive representations” to the benefits of the complete in-domain continued-training procedure.
2. Clarify why the memory is called temporal when the described inputs are clip-level embeddings, how class-specific memory slots are used at inference, how mixup and cutmix are applied consistently across the three modalities, and whether the V-JEPA predictor contributes during Stage 2.

---

> ### Author Response · Authors · 2026-07-22
> **Response to Reviewer wpyQ**
>
> We thank the reviewer for the careful and direct feedback. We agree that several original claims were stronger than the evidence supported. The revision therefore narrows the interpretation and adds controls to better separate Stage 1 adaptation, AMHF fusion, and modality contribution.
>
> ### 1. Stage 1 adaptation and claim scope
>
> We revised the Stage 1 claim so that it no longer implies that predictive latent learning is universally superior to other self-supervised objectives. The paper now makes the narrower claim that V-JEPA2-style in-domain adaptation is useful within the complete Emotion-JEPA pipeline. We added a compute-bounded contrastive adaptation control: in the full AMHF pipeline, public V-JEPA2 without Stage 1 obtains 77.80% WAF, contrastive-adapted V-JEPA2 obtains 78.28%, and predictive-adapted Emotion-JEPA obtains 85.72%. We explicitly state that the contrastive run is a bounded control, not an exhaustive comparison of self-supervised objectives.
>
> **Where addressed:** Section 4.7 and Table 6; Section 4.9; Appendix B.
>
> ### 2. Visual-only diagnostics
>
> We added frozen-probe visual diagnostics before and after adaptation. These show only a small difference between public V-JEPA2 and predictive-adapted V-JEPA2, so the paper no longer treats frozen probing as standalone evidence of universally better visual representations. The main evidence for Stage 1 is now framed as its effect inside the full multimodal pipeline.
>
> **Where addressed:** Section 4.7 and Table 6.
>
> ### 3. Reliability and uncertainty wording
>
> We revised the AMHF terminology throughout the paper. The learned AMHF quantities are now described as confidence-normalized interaction weights or fusion gates, not calibrated uncertainty estimates. We also added missing/noisy/shuffled modality diagnostics and interpret them as modality-dependence stress tests rather than uncertainty calibration evidence.
>
> **Where addressed:** Section 3.5; Section 4.5; Section 4.9; Appendix F.
>
> ### 4. Fusion controls, capacity, and complexity
>
> We removed the implication that cross-attention is a fully capacity-matched baseline. The fusion results are now described as controlled comparisons using the same encoders, split, projection dimension, and evaluation protocol. We added concat-MLP, cross-attention, simple audio-gated fusion, and modulator variants. The audio-modulated AMHF model reaches 85.72% WAF, compared with 80.31% for concat-MLP, 80.56% for cross-attention, 81.92% for simple audio-gated fusion, 81.09% for video-modulated AMHF, and 82.97% for text-modulated AMHF. We also added parameter and compute reporting to make the trade-off clearer.
>
> **Where addressed:** Section 4.3 and Table 2; Section 4.8; Appendix H.
>
> ### 5. Modality contribution
>
> We added independently trained unimodal baselines and separated them from auxiliary heads. The independent audio-only model reaches 81.13% WAF, the independent text-only baseline reaches 36.88% WAF, and the fused Emotion-JEPA model reaches 85.72% WAF. This makes the modality evidence more direct and avoids interpreting jointly trained auxiliary heads as true unimodal upper bounds. Corruption diagnostics further show how performance changes when individual modalities are degraded.
>
> **Where addressed:** Section 4.5 and Table 4; Appendix F.
>
> ### 6. Method clarifications
>
> We clarified that the V-JEPA predictor and EMA target encoder are used only in Stage 1 and discarded before Stage 2. The revised method also specifies the latent MAE/L1 Stage 1 loss, AMHF memory slots, auxiliary objective, trainable components, inference flow, and mixup/CutMix behavior. Mixup/CutMix are now described as video-stream augmentations only: audio and text remain paired with the original sample, while fused and auxiliary losses use the same mixed-label criterion. AMHF memory slots are described as learned refinement parameters, not class-specific labels used at inference. We also clarified that the component previously referred to as temporal memory does not maintain recurrent state across frames, clips, or batches; in the submitted AMHF configuration, it is applied after temporal pooling and is therefore described as memory-based refinement using learned prototype slots.
>
> **Where addressed:** Sections 3.2, 3.5, and 3.6; Appendices B, C, and H.
>
> ### 7. Broader impact
>
> We expanded the limitations to state that emotion labels are noisy task annotations rather than definitive measurements of internal state. We also added responsible-use discussion covering consent, privacy, demographic and cultural bias evaluation, and avoidance of high-stakes use.
>
> **Where addressed:** Section 5.
>
> Overall, the revised manuscript makes a narrower claim: Emotion-JEPA is a controlled study of in-domain visual adaptation and structured audio-modulated fusion, without claiming universal predictive-objective superiority, calibrated uncertainty, capacity-matched superiority over cross-attention, or strong cross-dataset generalization beyond the evidence provided.

---

### Author Response · Authors · 2026-07-22
**Author - Revision Summary**

We thank the reviewers for the careful feedback. We have uploaded a revised manuscript that narrows the claims, adds controlled evidence, and clarifies the implementation. Emotion-JEPA is now framed as a controlled study of affect-domain visual adaptation and audio-modulated fusion for MER, rather than as a pseudo-labeling or leaderboard-optimization system.

**1. Clarified contribution positioning and narrowed claims.**

First, we clarified the contribution and comparison scope. The JEPA objective, student-target training structure, and masked latent prediction setup are now described as inherited from V-JEPA/V-JEPA2. Our contribution is applying V-JEPA2-style in-domain adaptation to MER and combining it with AMHF under a no-pseudo-labeling protocol. We also revised the comparison text so that the paper does not claim SOTA performance over pseudo-labeling, voting, or self-training systems. The revised Table 1 explicitly places Emotion-JEPA relative to both stronger pseudo-labeling systems and the MERTools no-pseudo baseline.
**Where addressed:** Introduction; Related Work; Section 4.2; Table 1.

**2. Strengthened empirical support for AMHF through fusion controls, modulator variants, and corruption diagnostics.**
Second, we added stronger controls for AMHF and modality contribution. The revision includes concat-MLP late fusion, cross-attention fusion, simple audio-gated fusion, video/text-modulated AMHF variants, independent unimodal baselines, and modality corruption diagnostics. In the fusion comparison, the canonical audio-modulated AMHF model obtains 85.72 WAF, compared with 80.31 for concat-MLP, 80.56 for cross-attention, 81.92 for simple audio-gated fusion, 81.09 for video-modulated AMHF, and 82.97 for text-modulated AMHF. These experiments separate the fused model from auxiliary heads and show that AMHF is not only benefiting from the shared encoders or projection space.
**Where addressed:** Section 4.3; Table 2.

**3. Added independent unimodal baselines**
Third, we added independent modality evidence. The revised paper distinguishes independently trained unimodal models from auxiliary heads inside the jointly trained system. The independent audio-only model is strong at 81.13 WAF, while the full multimodal model reaches 85.72 WAF. We also added modality corruption diagnostics by zeroing, noising, or shuffling modalities. These are now presented as modality-dependence stress tests rather than as calibration evidence.
**Where addressed:** Section 4.5; Table 4; Appendix F.

**4. Revised “reliability/uncertainty” language to soften claims**

Fourth, we corrected the reliability/uncertainty language. AMHF weights are now described as learned confidence-normalized fusion gates, not calibrated uncertainty estimates. The revised text states that these gates are learned through the supervised objective and should be interpreted as interaction weights rather than probabilistic uncertainty. We also added sensitivity analysis over the number of experts and top-k routing choices, and avoid claiming human-interpretable expert specialization beyond the evidence provided.
**Where addressed:** Sections 3.5, 4.6, and 4.9; Table 5; Appendix C.4.

**5. Added visual adaptation diagnostics**
Fifth, we narrowed the Stage 1 interpretation. Frozen-probe and full-pipeline diagnostics now compare public V-JEPA2, predictive-adapted V-JEPA2, and a compute-bounded contrastive adaptation control. The frozen-probe results show only a small standalone visual difference, so the revised claim is limited to gains within the full Emotion-JEPA pipeline, not universal visual-representation superiority. In the full AMHF pipeline, public V-JEPA2 without Stage 1 obtains 77.80 WAF, the contrastive adaptation control obtains 78.28 WAF, and predictive-adapted Emotion-JEPA obtains 85.72 WAF.
**Where addressed:** Section 4.7; Table 6; Appendix B.

**6. Added AMHF routing and modulator sensitivity analyses.**

Finally, we improved reproducibility and limitations. The revised method specifies Stage 1/Stage 2 trainable components, predictor and EMA-teacher use, latent MAE Stage 1 loss, AMHF memory size, EATP/text data flow, parameter counts, compute cost, MER2025 transfer limitations, and responsible-use concerns. The paper now explicitly treats cross-dataset generalization, large-model cost, face-centric preprocessing, audio dependence, synchronization assumptions, and emotion-label ambiguity as limitations.
**Where addressed:** Sections 3, 4.8, 4.9, and 5; Appendices B, C, E, and H.

Overall, the revised manuscript now presents Emotion-JEPA as a controlled study of affect-domain visual adaptation and structured audio-modulated fusion for MER, rather than as a claim of universal predictive-learning superiority, calibrated uncertainty estimation, or state-of-the-art leaderboard performance. We hope these revisions address the reviewers’ concerns and make the contribution, evidence, and limitations substantially clearer.